# hu.MAP3.0: atlas of human protein complexes by integration of >25,000 proteomic experiments

Samantha N Fischer[1,5], Erin R Claussen[1,5], Savvas Kourtis[2], Sara Sdelci [2], Sandra Orchard[3], Henning Hermjakob [3], Georg Kustatscher [4] & Kevin Drew [1✉]

## Abstract

**Macromolecular protein complexes carry out most cellular functions. Unfortunately, we lack the subunit composition for many human protein complexes. To address this gap we integrated >25,000 mass spectrometry experiments using a machine learning approach to identify >15,000 human protein complexes. We show our map of protein complexes is highly accurate and more comprehensive than previous maps, placing nearly 70% of human proteins into their physical contexts. We globally characterize our complexes using mass spectrometry based protein covariation data (ProteomeHD.2) and identify covarying complexes suggesting common functional associations. hu.MAP3.0 generates testable functional hypotheses for 472 uncharacterized proteins which we support using AlphaFold modeling. Additionally, we use AlphaFold modeling to identify 5871 mutually exclusive proteins in hu.MAP3.0 complexes suggesting complexes serve different functional roles depending on their subunit composition. We identify expression as the primary way cells and organisms relieve the conflict of mutually exclusive subunits. Finally, we import our complexes to EMBL-EBI's Complex Portal (https://www.ebi.ac.uk/complexportal/home) and provide complexes through our hu.MAP3.0 web interface (https://humap3.proteincomplexes.org/). We expect our resource to be highly impactful to the broader research community.**

**Keywords** Protein Complex; Protein Interaction; Mutually Exclusive; Disease Candidates; Machine Learning
**Subject Categories** Computational Biology; Proteomics

## Introduction

Proteins are the functional units of the cell yet they carry out cellular functions by self assembling into large macromolecular protein complexes. To gain a more complete understanding of human cells, it is essential to identify and characterize these protein complexes. Major advances in high throughput proteomics and machine learning

workflows have allowed both the collection of large datasets and the integration of those datasets into more accurate and complete sets of protein complexes. In particular, experimental workflows based on affinity purification mass spectrometry (AP-MS) (Huttlin et al, 2021, 2017, 2015; Hein et al, 2015; Boldt et al, 2016; Cho et al, 2022; Malovannaya et al, 2011), co-fractionation mass spectrometry (CF-MS) (Wan et al, 2015; McWhite et al, 2020; Cox et al, 2024; Skinnider et al, 2021; Havugimana et al, 2012; Kirkwood et al, 2013), and proximity labeling (Youn et al, 2019; Gupta et al, 2015) have been applied to different cells, tissues, and organisms generating a large compendium of proteomics data focused on identifying protein complexes. We have previously integrated several of these datasets using machine learning in our hu.MAP1.0 and hu.MAP2.0 resources which have been extremely valuable at identifying candidate disease genes, functionally annotating the proteome's least characterized proteins, and identifying multifunctional promiscuous proteins (Drew et al, 2021, 2017a). In total hu.MAP2.0 placed half of the human proteins (~10,000) into protein complexes (Drew et al, 2021).

Orthogonal approaches, based on co-expression/covariation of proteins in mass spectrometry-based datasets across many conditions, have been developed to identify functional modules of the cell (Trip et al, 2025; Kustatscher et al, 2019, 2023). While these networks are not physical in nature, the efforts have shown subunits of protein complexes are often co-expressed and provide a powerful signature for shared co-complex interactions.

Here we describe an improved hu.MAP3.0 computational workflow for the integration of high throughput proteomics experiments to identify human protein complexes. Using this workflow we integrate >25,000 mass spectrometry experiments from multiple approaches (i.e., AP-MS, CF-MS, proximity labeling) and identify greater than 15,000 protein complexes for ~70% of the human proteome (>13,700 total proteins). We compare physical complexes identified by hu.MAP3.0 to protein covariation networks and find many complexes have tight covariation patterns. Additionally, we utilize AlphaFold structural models of protein pairs to identify mutually exclusive proteins (i.e., pairs of proteins that cannot coexist in complex due to occupying the same binding interface). These are further supported by covariation networks. In total we identify 5871 mutually exclusive pairs which point to potential regulation and functional outcomes of complexes in different cell types, tissues, and organelles. We also

[1]Department of Biological Sciences, University of Illinois at Chicago, Chicago, IL 60607, USA. [2]Centre for Genomic Regulation (CRG), The Barcelona Institute of Science and Technology, Barcelona, Spain. [3]European Molecular Biology Laboratory, European Bioinformatics Institute (EMBL-EBI), Wellcome Genome Campus, Hinxton, Cambridge CB10 1SD, UK. [4]Centre for Cell Biology, University of Edinburgh, Edinburgh EH9 3BF, UK. [5]These authors contributed equally: Samantha N Fischer, Erin R Claussen.
✉E-mail: ksdrew@uic.edu

analyze our map to identify understudied proteins which physically associate with well-annotated complexes. We place 472 understudied proteins into complexes, several of which are disease associated. This provides testable hypotheses as to their function. Finally, we make our identified complexes available through two major resources, a hu.MAP3.0 web resource (https://humap3.proteincomplexes.org) and EMBL-EBI's Complex Portal (https://www.ebi.ac.uk/complexportal/home) (Balu et al, 2025).

# Results

To construct a more accurate and comprehensive set of protein complexes, we use machine learning to integrate published high throughput proteomics experiments. Each experimental dataset targets interactions from different cell types and different parts of the proteome. This therefore provides a broad sampling of human protein complexes as well as multiple lines of evidence for each protein complex. In Fig. 1A, we show a schematic representation of the full machine learning pipeline to produce protein complexes. The pipeline consists of 3 major steps: (1) organization and generation of features predictive of protein interactions, (2) training of the machine learning model to generate a protein interaction network, and (3) clustering of the network to identify complexes. This final step results in 15,326 complexes covering 13,769 human proteins and consisting of 159,451 total scored protein interactions. This set of complexes, which we call the hu.MAP3.0 human protein complex map, considerably increases coverage compared to previous complex maps (Fig. 1B–D).

## Integration of high throughput proteomics datasets assembles evidence of protein interactions

We first describe our development of the hu.MAP3.0 machine learning pipeline. The initial phase includes organizing evidence features from experimental datasets from CF-MS (Wan et al, 2015), AP-MS (Huttlin et al, 2021, 2017, 2015; Hein et al, 2015; Boldt et al, 2016; Malovannaya et al, 2011), Proximity Labeling (Youn et al, 2019; Gupta et al, 2015), and RNA pulldown experiments (Treiber et al, 2017). In our current work, we include a significant number of new experiments from the BioPlex3 (Huttlin et al, 2021) effort which adds ~10k new AP-MS experiments to our workflow. Evidence for each protein interaction from all experiments were collated into a feature matrix. CF-MS features represent the similarity of a pair of protein's elution in a biochemical separation with high similarity suggesting an interaction. Pulldown and proximity features contain metrics of the presence of an individual prey protein identified in a bait protein experiment. These measures provide models for bait-prey interactions. To capture additional evidence of interaction between prey proteins, we reanalyzed AP-MS, RNA pulldown, and Proximity Labeling datasets using our weighted matrix model approach (WMM) (Drew et al, 2017a; Hart et al, 2007) which uses the hypergeometric test to identify pairs of proteins seen in these large datasets more often than random (see Methods). This feature has been extremely valuable at identifying novel interactions in previous work (Drew et al, 2021, 2017a).

## Construction of a machine learning model to identify protein interactions

We next build a machine learning model using evidence features to identify pairs of proteins that interact in a complex. Once we constructed our feature matrix, we label the pairs of proteins with our training set derived from manually curated protein complexes in the Complex Portal (Balu et al, 2025) (see Methods). We then use the feature matrix labeled with training data to build a machine learning classifier. For this step we use AutoGluon (Erickson et al, 2020), which automates model and hyper-parameter selection across 13 machine learning models. The AutoGluon step builds a weighted ensemble model based on the top performing classifiers. We then generate a network consisting of ~26 million protein pairs where all pairs of proteins are classified using the final weighted ensemble model. The model produces a confidence score for each pair of proteins representing the estimated probability of the pair interacting based on the evidence.

We then evaluate our final model using a precision recall analysis evaluated on a leave out test set of protein interactions (Fig. 2A). We compare our current hu.MAP3.0 protein interaction network against our prior protein interaction network, hu.MAP2.0, as well as the Bioplex3 interaction network. We see our current hu.MAP3.0 outperforms the other networks, increasing in both precision and recall, with an area under the precision recall curve (AUPRC) of 0.68. Bioplex3 and hu.MAP2.0 have an AUPRC of 0.38 and 0.61, respectively. To assess how well the confidence score generated by our model reflects the true precision of the network, we compare the confidence score to precision as determined by the leave out test set. In Fig. EV1A, we see the confidence score closely reflects the true precision. We also see the relationship between confidence score and test set label (true or false interaction) in Fig. EV1B. Here we see the majority of true interactions have substantially higher confidence scores and the vast majority of false interactions have very low confidence scores. To assess if the model is well-calibrated, we looked at the interactions containing the ~4000 newly added proteins to hu.MAP3.0 (Fig. EV1C). We see that the confidence scores for pairs containing these proteins reflect the background distribution of all pairwise scores, indicating the stability of the model's predictions.

## Feature importance

We next evaluated individual features to estimate their importance to our model. Using AutoGluon's feature importance module (see Methods), we observe the majority of top performing individual features are those from our derived weighted-matrix model features, AP-MS, and CF-MS (Fig. EV1D, Dataset EV1). To better discriminate the predictive value of specific data types as whole, we then evaluated features based on experiment class/method, in particular AP-MS experiments, CF-MS experiments, and weighted-matrix model (WMM) features. We ablated each set of features and evaluated them using precision-recall (PR) curve analysis (Fig. EV1E). We observe the ablation of both AP-MS and WMM features leads to performance loss, while ablation of CF-MS features results in limited performance loss. To test the degree of CF-MS signal in our model, we ablated all features except the CF-MS features (Fig. EV1E). We find CF-MS features do provide

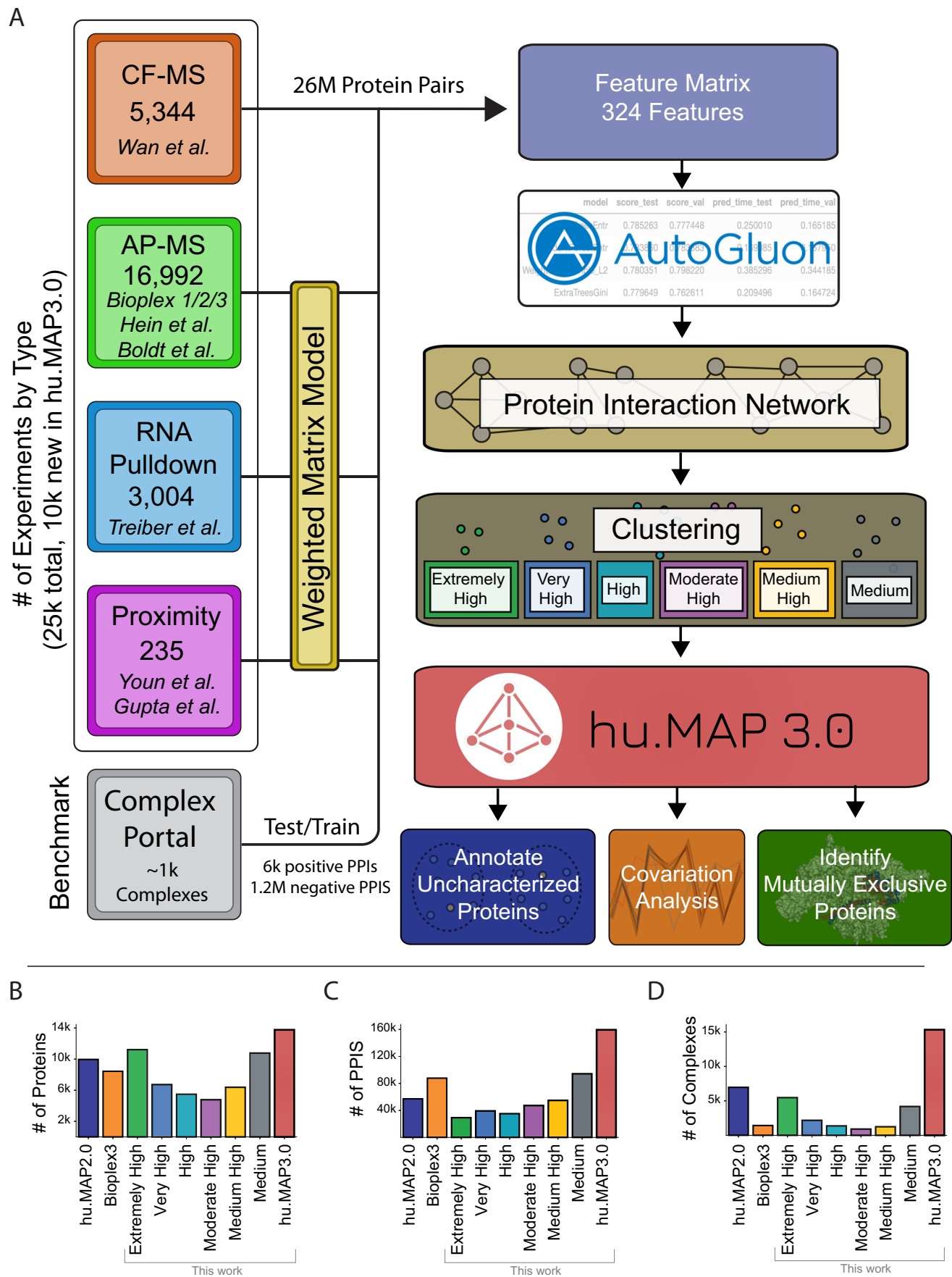

◀  **Figure 1.  Machine learning workflow identifies human protein complexes.**

(A) Representation of the hu.MAP3.0 workflow which integrates ~25k mass spectrometry experiments to identify protein complexes. A feature matrix for machine learning classification of protein pairs was constructed using four classes of experimental datatypes (i.e., Cofractionation Mass Spectrometry (CF-MS), Affinity Purification Mass Spectrometry (AP-MS), Weighted Matrix Model applied to RNA Pulldown Mass Spectrometry experiments, and Proximity Labeling Mass Spectrometry experiments). A reanalysis of AP-MS and proximity labeling data was performed using a Weighted Matrix Model. The feature matrix was labeled using known Complex Portal interactions and a classifier was trained using the AutoGluon model selection pipeline. A protein interaction network was built by applying the model to ~26 million pairs of proteins that provided a confidence score of co-complex interaction. The protein interaction network was clustered producing complexes of "Extremely High" confidence to "Medium" confidence. Complexes were then analyzed to annotate uncharacterized proteins and identify mutually exclusive proteins. (B) Comparison of human proteome coverage between hu.MAP2.0 (dark blue), Bioplex3 (orange), hu.MAP3.0 (red), and the 6 levels of complex confidence of hu.MAP3.0 (green = "Extremely High", blue = "Very High", aqua = "High", purple = "Moderately High", yellow = "Medium High", and gray = "Medium"). The complete set of hu.MAP3.0 complexes covers 13,769 human proteins outperforming hu.MAP2.0 and Bioplex3. (C) Comparison of number of protein interactions (PPIs) across hu.MAP2.0, Bioplex3, hu.MAP3.0, and 6 levels of complex confidences. Same colors as in (B). The hu.MAP3.0 complex map considerably increases the number of total protein interactions including 159,451 total scored interactions. (D) Comparison of number of complexes in each dataset. Same colors as in (B). hu.MAP3.0 contains 15,326 complexes, a substantial increase in the number of complexes than previously published protein complex maps.

predictive signal, although reduced from other features. This suggests high quality AP-MS and other experiments have substantial coverage of test set interactions and our model downweights CF-MS features.

## Identification of protein complexes within protein interaction network

We next cluster the full protein interaction network to identify protein complexes. This is done using a two stage clustering approach (Fig. 2B, see Methods). To determine optimal clusters, we test combinations of clustering parameters and confidence thresholds for the input protein interaction network. We use each parameter combination to generate a set of clusters which are then evaluated using the $k$-clique precision and recall measure which globally compares a set of clusters against a training set of complexes (Drew et al, 2017a). Figure 2C shows the evaluation of all sets of clusters from 1080 parameter combinations on the set of curated training complexes. In this evaluation (Fig. 2C), we see a tradeoff between precision and recall, where some cluster sets have high precision and low recall, while other cluster sets have lower precision and high recall. We select six clustering sets along this boundary to combine into our final set of hu.MAP3.0 predicted complexes. We rank each set of clusters according to their precision value, considering the highest precision as "Extremely High" confident, followed by "Very High", "High", "Moderately High", "Medium High", and "Medium". The last of which, "Medium" has the highest recall of all other cluster sets. The full set of hu.MAP3.0 complexes can be found in Dataset EV2. We then evaluate all six cluster sets and their union on a benchmark of curated leave out test complexes (Fig. 2D). We see that all six cluster sets have a consistent order in terms of their tradeoff between precision and recall, with "Extremely High" clusters having the highest precision and lowest recall, and the "Medium" clusters having the highest recall and lowest precision. Specific complexes were seen multiple times throughout the confidence values. We therefore assigned each final complex the highest confidence level in which the complex was seen.

Figure EV2A shows the distribution of complex size relative to complex confidence levels. We see complex size increase with lower levels of confidence (i.e., higher recall). To illustrate this further, we inspected variants of the MCM complex (Fig. EV2B). Variants of the MCM complex in our Extremely High confidence clusters include huMAP3_02602.1 which contains the dimer of MCM4 and MCM6 while our next highest confidence clusters (i.e., Very High) contains a complex with the majority of the subunits from the entire MCM complex (huMAP3_05529.1). Finally, in our Medium confidence clusters we identify the MCM complex associated with the GINS complex, which is known to physically interact, and other DNA replication factors (huMAP3_14758.1). This example demonstrates that our clustering strategy effectively identifies subcomplexes, complexes and supercomplexes with known biological and structural roles. We observe in Fig. EV2C that most proteins are present in multiple complexes. Our clustering method therefore generates a degree of redundancy where it attempts to capture subcomplexes or core members of complexes being confidently identified in high confidence levels and additional auxiliary subunits included in complexes with lower confidence values.

The total number of complexes is highest in our highest confidence level (Fig. 1D). Figure 1B shows our highest confidence level has broad coverage of the proteome. We observe the total number of protein–protein interactions (PPIs) per confidence level rise as we decrease in confidence, again suggesting that high confidence complexes are smaller representing core subunits (Fig. 1C).

We also evaluate the union of all cluster sets, and observe it has the highest recall over all the individual cluster sets as well as outperforms Bioplex3 clusters (Fig. 2D). Further, when evaluating our model pre- and post-clustering, we see that clustering marginally improves identification of true interactions (Fig. EV2D). We also compare the coverage of proteins, protein interactions, and complexes and see hu.MAP3.0 increases comprehensiveness over both Bioplex3 and hu.MAP2.0 (Fig. 1B–D). Notably, we observe >4800 complexes are enriched with annotations from Gene Ontology (GO) (Ashburner et al, 2000), KEGG (Kanehisa et al, 2014), Reactome (Fabregat et al, 2016), CORUM (Giurgiu et al, 2019), and/or Human Phenotype Ontology (HP) (Köhler et al, 2014) (Fig. EV2E, Dataset EV3). The size distribution of enriched complexes largely mirrors the distribution of all identified hu.MAP3.0 complexes indicating that annotation enrichment is not biased towards complex size (Fig. EV2F). We therefore expect unenriched complexes to have either novel functions or contain subunits yet to be annotated by annotation efforts. In addition, as a convenience we provide a non-redundant set of complexes (Dataset EV4, see Methods).

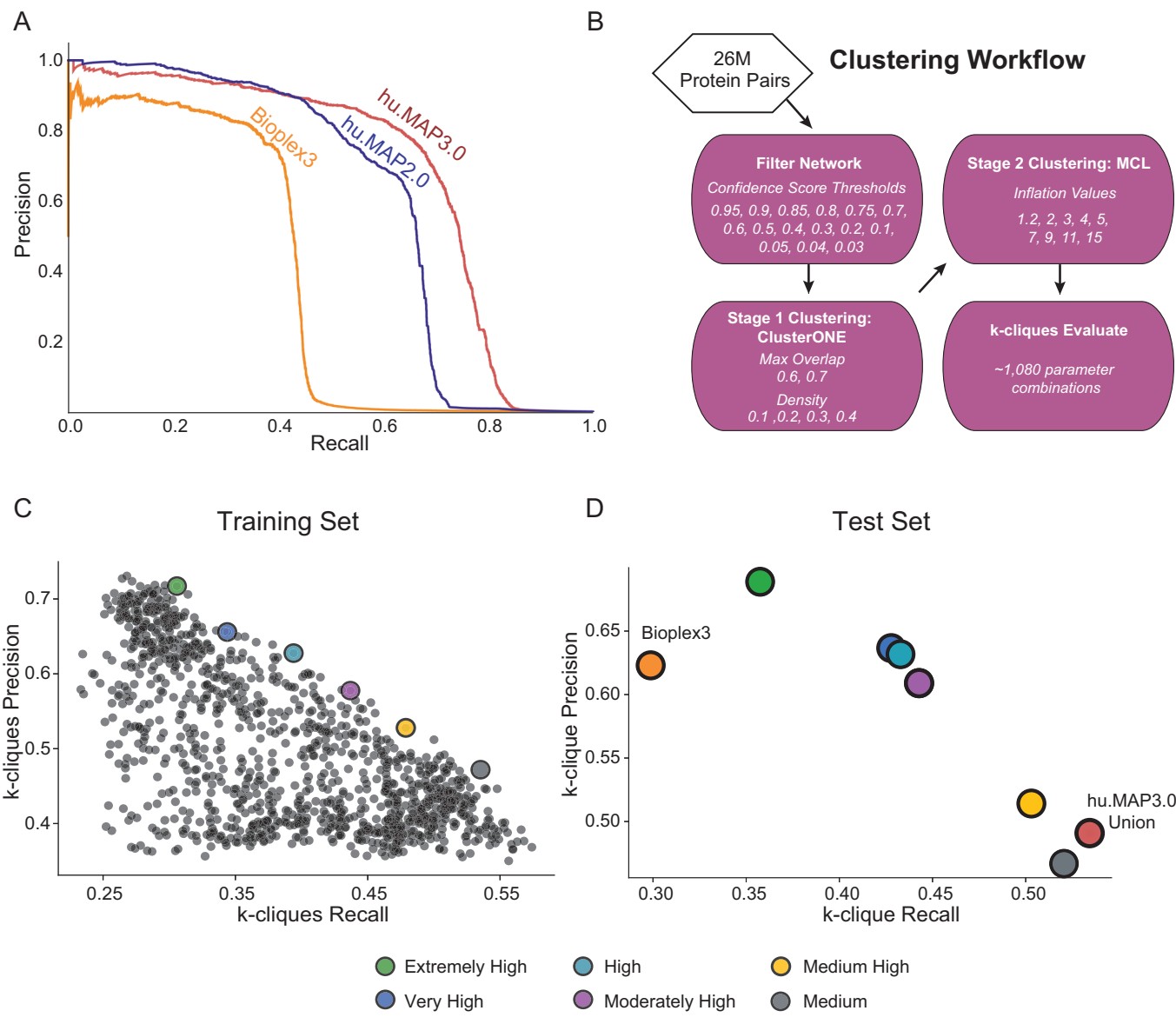

**Figure 2. hu.MAP3.0 outperforms previous complex maps.**

(A) Precision recall (PR) analysis on a test leave out set of gold standard co-complex interactions demonstrates hu.MAP3.0 is more accurate and comprehensive than alternative models, hu.MAP2.0 and Bioplex3. (B) Clustering workflow to identify optimal clustering parameters. (C) Scatterplot of *k*-clique precision recall measures for >1000 clustering parameters. *k*-clique evaluation in (C) is done on a training set of gold standard complexes to identify parameters that produce confident protein complexes. We observe a trade off between precision and recall. We selected six parameter sets that represented "extremely high" confidence (green) to "medium" confidence (gray) complexes. (D) *k*-clique evaluation of selected sets of complexes using a test leave out set of gold standard complexes. Consistent with (C), we observe the same tradeoff between precision and recall, in addition to maintaining ordering of sets of complexes. Additionally, the union of all hu.MAP3.0 complexes from the various confidence levels improves on recall. Extremely High, Very High, and High confident complexes improve in both precision and recall over Bioplex3 complexes.

## Protein covariation provides orthogonal evidence for hu.MAP3.0 complexes

Protein covariation analysis, also known as co-regulation analysis, identifies proteins exhibiting coordinated changes in abundance across biological conditions. We and others have previously shown that protein covariation analysis is a powerful approach to capture functional relationships between proteins and provide novel insights into proteome organization (Kustatscher et al, 2019, 2023; Messner et al, 2023; Leduc et al, 2022; Lapek et al,

2017; Wang et al, 2017; Kanonidis et al, 2016). For our protein covariation resource, ProteomeHD, we re-processed thousands of mass spectrometry runs that had been manually selected to reflect protein abundance changes in response to various biological perturbations (Kustatscher et al, 2019). For example, typical experimental designs included comparisons of mutant to wildtype cells, and drug *vs* control treatments. We then used a decision-tree-based algorithm to determine likely functional relationships between proteins based on correlated protein abundance patterns. Therefore, hu.MAP and ProteomeHD are

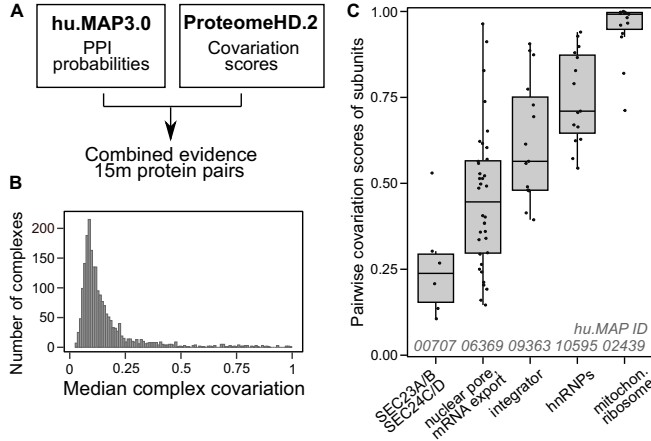

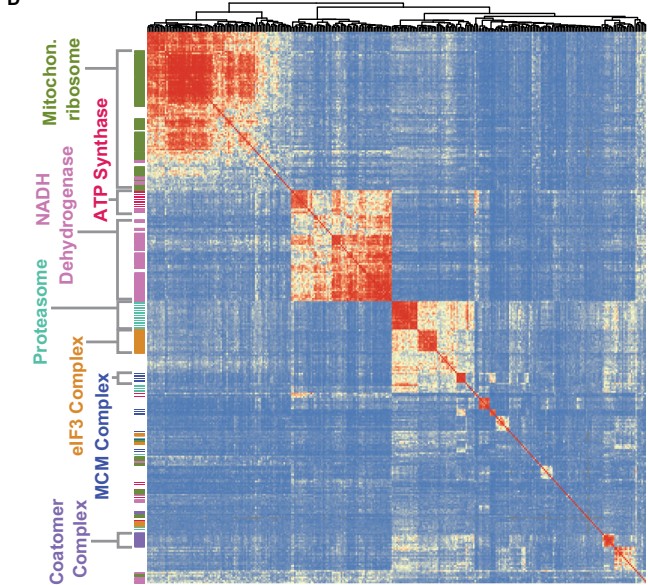

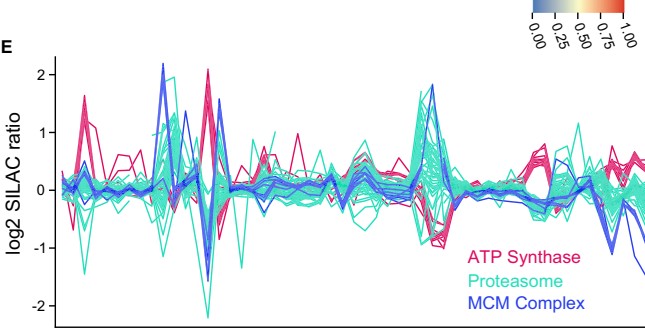

**Figure 3.   Integrating hu.MAP3.0 with covariation data reveals complementary insights into protein complex biology.**

(A) Workflow of pairwise protein–protein interaction probabilities (hu.MAP3.0) with pairwise protein covariation probabilities (ProteomeHD.2). Protein identifiers were combined at the protein-level, with only common pairs between datasets used for downstream analysis. (B) Histogram showing median covariation of subunits for hu.MAP3.0 complexes. Complexes with fewer than 4 subunits, a hu.MAP3.0 confidence score below 4, or less than 50% coverage in ProteomeHD.2 were excluded. (C) Boxplots of pairwise subunit covariation scores for five example complexes ($N = 6$, $N = 36$, $N = 15$, $N = 15$, $N = 15$, in order as plotted from left to right), selected to represent diverse degrees of complex-level covariation. Upper and lower bounds of each box correspond to quartiles of the distribution with first (Q1, 25th percentile) and third quartile (Q3, 75th percentile), respectively. The line in each box corresponds to the median. The whiskers extend to the minimum and maximum values within 1.5 × interquartile range (IQR) from Q1 and Q3. Individual data points are plotted. (D) Heatmap showing hierarchical clustering of protein covariation scores for proteins assigned to hu.MAP3.0 complexes. Subunits of selected complexes are highlighted as a rug plot on the left, indicating that many complex subunits cluster together. Proteins are plotted in the same order on both axes, revealing covariation between different complexes. Only complexes with 50% coverage in ProteomeHD.2 and median covariation score of 0.45 were included. (E) Log2 fold-changes (SILAC ratios) across a subset of perturbation experiments from ProteomeHD.2 for subunits belonging to three selected complexes.

functional relationships based on covariation rather than physical interaction.

Despite these differences, we and others have previously found that many strongly covarying proteins are in fact subunits of the same protein complex (Kustatscher et al, 2019, 2023; Lapek et al, 2017; McShane et al, 2016), suggesting that many complexes have strong covariation signatures. Therefore, we reasoned that protein covariation could provide orthogonal evidence to validate, enrich and annotate our human protein complex map. To test this we combined protein–protein interaction probabilities from hu.MAP3.0 with covariation scores from ProteomeHD.2 (ProHD.2), a recently developed successor of ProteomeHD (Fig. 3A). In comparison to the first iteration of ProteomeHD, ProHD.2 incorporates a larger number of biological conditions and uses an improved, supervised machine-learning strategy to determine proteins whose abundance covaries across these conditions (Kourtis, 2025) (see Methods). In total, 15 million protein pairs are covered by both hu.MAP3.0 and ProteomeHD.2 (Fig. EV3A). To assess if the datasets are indeed complementary, we compared them to high-quality, experimentally determined pairwise interactions from the OpenCell project (Cho et al, 2022), which neither hu.MAP3.0 nor ProHD.2 used as an input. Indeed, protein pairs with high association scores in both hu.MAP3.0 and ProHD.2 are much more likely to be captured by OpenCell than random pairs (29% vs 0.3%, respectively), and significantly more likely than pairs captured by only one of the two approaches (Fig. EV3B). This suggests that protein covariation provides orthogonal evidence to support protein interactions in hu.MAP3.0.

## Covariation provides complementary insights into complex biology

We next asked what the combined hu.MAP3.0 and ProHD.2 data could tell us about the nature of protein complex expression. We find that complexes show varying degrees of subunit covariation

conceptually similar. Both approaches integrate thousands of proteomics experiments using machine-learning to infer protein–protein associations in a data-driven manner. However, they are also highly complementary as they rely on fundamentally different types of evidence and raw data. While hu.MAP integrates experiments that determine physical interactions, protein covariation analysis draws on protein abundance data from an entirely distinct set of mass spectrometry experiments, highlighting

(Fig. 3B). Some hu.MAP3.0 protein complexes are tightly covarying modules with all subunits showing highly coordinated abundance changes, such as the mitochondrial ribosome and a nuclear ribonucleoprotein complex (Fig. 3C). Other complexes, such as the nuclear pore and integrator complexes, have weaker but still very clear covariation signatures. However, many complexes show low subunit covariation. In fact overall correlation between hu.MAP3.0 confidence scores and ProHD.2 covariation scores is low (Pearson correlation coefficient = 0.29, see Methods). There could be technical explanations for this, such as a failure of ProHD.2 to capture covariation patterns for some proteins (false negatives in ProHD.2) as well as potential complex prediction errors (false positives in hu.MAP3.0). However, we find that the degree of complex covariation correlates only mildly with hu.MAP3.0 confidence levels (Appendix Fig. S1) and correlation between average hu.MAP3.0 confidence scores and average ProtHD.2 scores per complex is low (Pearson correlation coefficient = 0.1, see Methods). This suggests that poor subunit covariation is not simply the result of potential complex prediction errors. As described previously, low covariation scores may also be observed for biological reasons, including complex subunits having additional, non-overlapping functions outside of the complex, and an increase of protein subunit abundance to compensate for a weak affinity protein interaction (Matalon et al, 2014). Another potential reason for low subunit covariation is exemplified by a hu.MAP3.0 complex consisting of SEC23A, SEC23B, SEC24C, and SEC24D (Fig. 3C). SEC23A/SEC23B and SEC24C/SEC24D are paralogues, respectively (Fromme et al, 2008), which could potentially form mutually exclusive SEC23-24 dimers. We explore the possibility to use our data to systematically detect mutually exclusive subunits below. Taken together, these results suggest that, while a high covariation score supports physical interaction evidence, a low covariation score is more difficult to interpret and does not necessarily imply a lack of physical interaction.

## Protein covariation captures associations between and within complexes

Unlike traditional protein–protein interaction mapping, protein covariation can identify functionally associated proteins even when they do not physically interact. This is because proteins involved in related biological processes may exhibit coordinated abundance changes despite the lack of direct physical contact. For example, we have shown that the peroxisomal protein PEX11β covaries with proteins involved in mitochondrial respiration (Kustatscher et al, 2019). Now, we leverage this property to assess inter-complex covariation, determining which complexes display similar expression patterns across the ProHD.2 dataset. This analysis enabled us to construct a map connecting dozens of hu.MAP3.0 complexes (Fig. 3D). It identifies associations between functionally related complexes, such as the mitochondrial respiratory complexes NADH dehydrogenase (complex I) and ATP synthase (complex V). However, these two complexes do not covary with the mitochondrial ribosome, indicating that the detection of inter-complex associations is function-specific and does not merely capture subcellular localization. We observe strong covariation between two hu.MAP3.0 complexes enriched in translation initiation factors and proteasomal proteins, respectively. This association may reflect a role of the proteasome in

translation quality control (Inada and Beckmann, 2024). However, some inter-complex associations are more difficult to interpret. For example, the proteasome shares a broadly similar expression profile with subunits of the MCM replication factor complex (Fig. 3D,E).

We next explored if covariation analysis could reveal substructures within individual protein complexes, using the proteasome as a well-annotated example. The 26S proteasome is a large 30+ subunit protein complex responsible for protein degradation and maintaining protein homeostasis. The full complex is made up of defined subcomplexes including a 20S core and 19S regulatory particles (Abi Habib et al, 2022). The hu.MAP3.0 complex huMAP3_09656.1 includes these core and regulatory subunits of the proteasome, along with tissue-specific core subunits and several assembly chaperones, activator complexes and other proteins known to associate with the proteasome. The degree of covariation between members of this complex reflects known biological properties (Fig. 4A). Subunits from the 20S core and the 19S regulatory particle exhibit strong covariation within their respective subcomplexes, with slightly lower covariation observed between these two substructures. The immunoproteasome-specific subunits PSMB8-10, which replace PSMB5-7 in immune cells, show weaker covariation with other core subunits compared to their canonical counterparts (Abi Habib et al, 2022). The two subunits of the PA28αβ activator (PSME1, PSME2) strongly covary with each other and with multiple proteasome subunits, but demonstrate weak covariation with the PA28γ activator (PSME3), which, despite structural similarity, has a distinct biological role (Abi Habib et al, 2022). Additionally, the proteasome assembly chaperone PAC3 (PSMG3), which forms a heterodimer with PAC4 (PSMG4), interacts preferentially with the α5 core subunit (PSMA5), as reflected in the higher covariation scores with these proteins (Satoh et al, 2019). Furthermore, covariation data support established proteasome interactions with other proteins, such as AKIRIN2, which plays a role in nuclear import of proteasomes (de Almeida et al, 2021), and UCHL5, a proteasome-associated deubiquitinase (Deol et al, 2020).

## Uncharacterized protein REX1BD interacts with proteasome lid subunits

Within the proteasome hu.MAP3.0 complex, we identify potential members not previously known to associate with the full complex. Specifically, we identified REX1BD, an uncharacterized protein as a high confidence co-complex interactor of the 19S proteasome lid, including 19S lid subunits PSMD13 (hu.MAP3.0 score = 0.99), PSMD14 (0.99), and PSMD7 (0.99). We also find evidence for physical association between PSMD14 and REX1BD through immunoprecipitation studies (Sowa et al, 2009). Figure 4B shows the evidence from individual proteomic experiments and analysis for REX1BD-PSMD13 interaction which included both pulldowns and weighted matrix models of the pulldown compendium. Additionally, we find REX1BD and PSMD13 demonstrate coordinated abundance changes in ProteomeHD.2 (Fig. 4A). Their covariation score of 0.43 places the pair in the top 0.6% of all associations scored by ProHD.2. To provide further evidence for this interaction, we used AlphaFold3 to model the three-dimensional structure of REX1BD and PSMD13. AlphaFold3 produced a confident model of the two proteins (pTM = 0.75) including a confident interaction interface (ipTM = 0.67) (Fig. 4C,D). These results suggest a potential role of REX1BD in the proteasome degradation pathway.

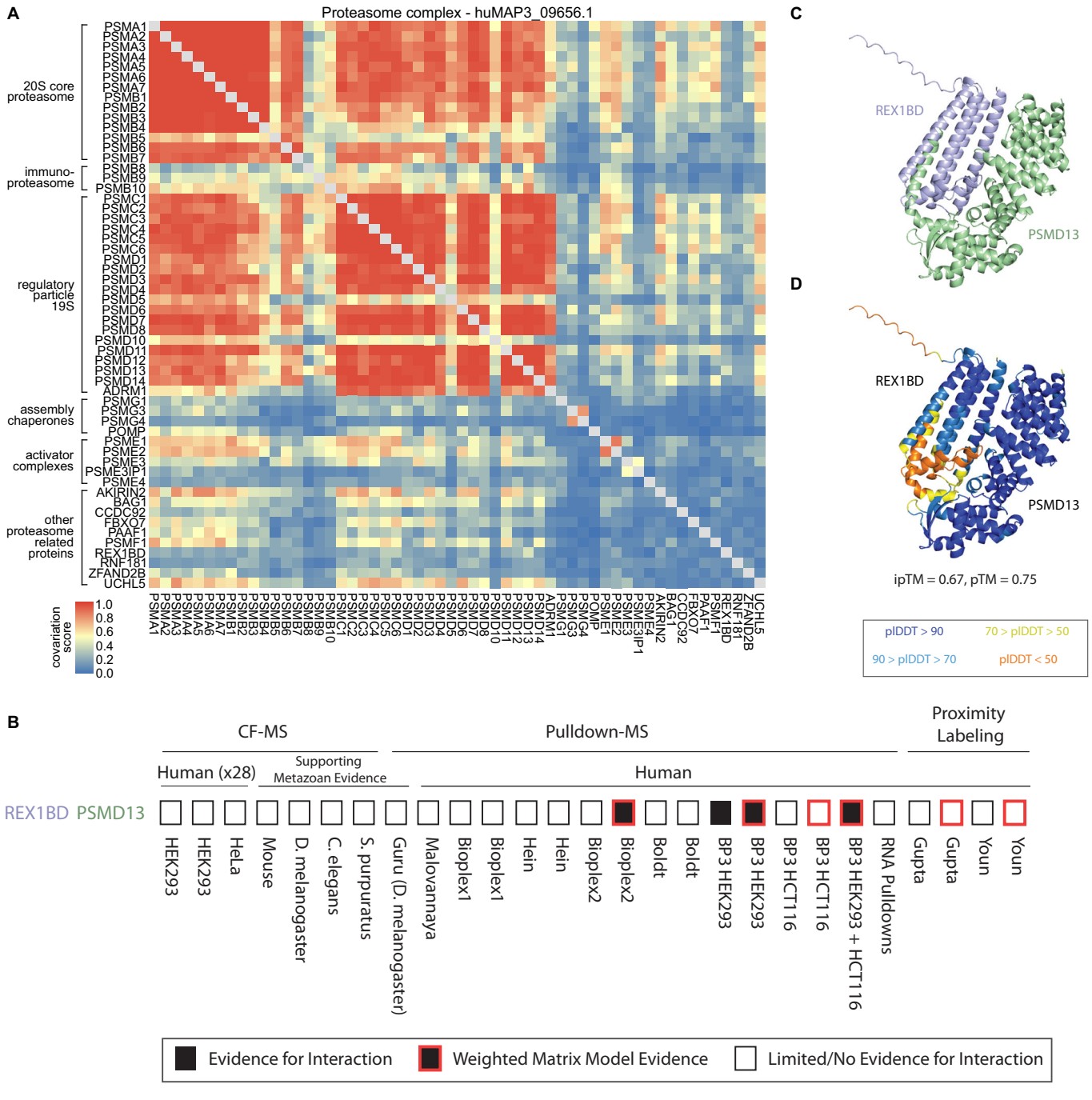

**Figure 4. Covariation identifies modules of proteasome and novel interactor of 19S regulatory lid.**

(A) Proteasome heatmap shows pairwise covariation score. (B) Graphical representation of evidence for protein interaction. (C) AlphaFold3 model of uncharacterized protein REX1BD (blue) and PSMD13 (green) subunit of proteasome lid. (D) Same model in (C), colored by pLDDT confidence score. Color scale for pLDDT is from blue (very high confidence) to orange (very low confidence).

## hu.MAP3.0 provides testable hypotheses to understudied proteins

Building upon our identification of the uncharacterized REX1BD as interacting with the 19S proteasome, we next searched for functional hypotheses for other uncharacterized proteins. Thousands of human proteins, many from essential genes and of biomedical importance,

remain understudied (Kustatscher et al, 2022). We and others have used protein complex maps as an unbiased way of transferring function annotations to uncharacterized proteins as well as identifying novel disease genes (Drew et al, 2021, 2017a; Gavin et al, 2002; Ho et al, 2002; Wang and Marcotte, 2010).

As an example of identifying novel disease genes, we find a novel interaction between disease gene IER3IP1 and an uncharacterized

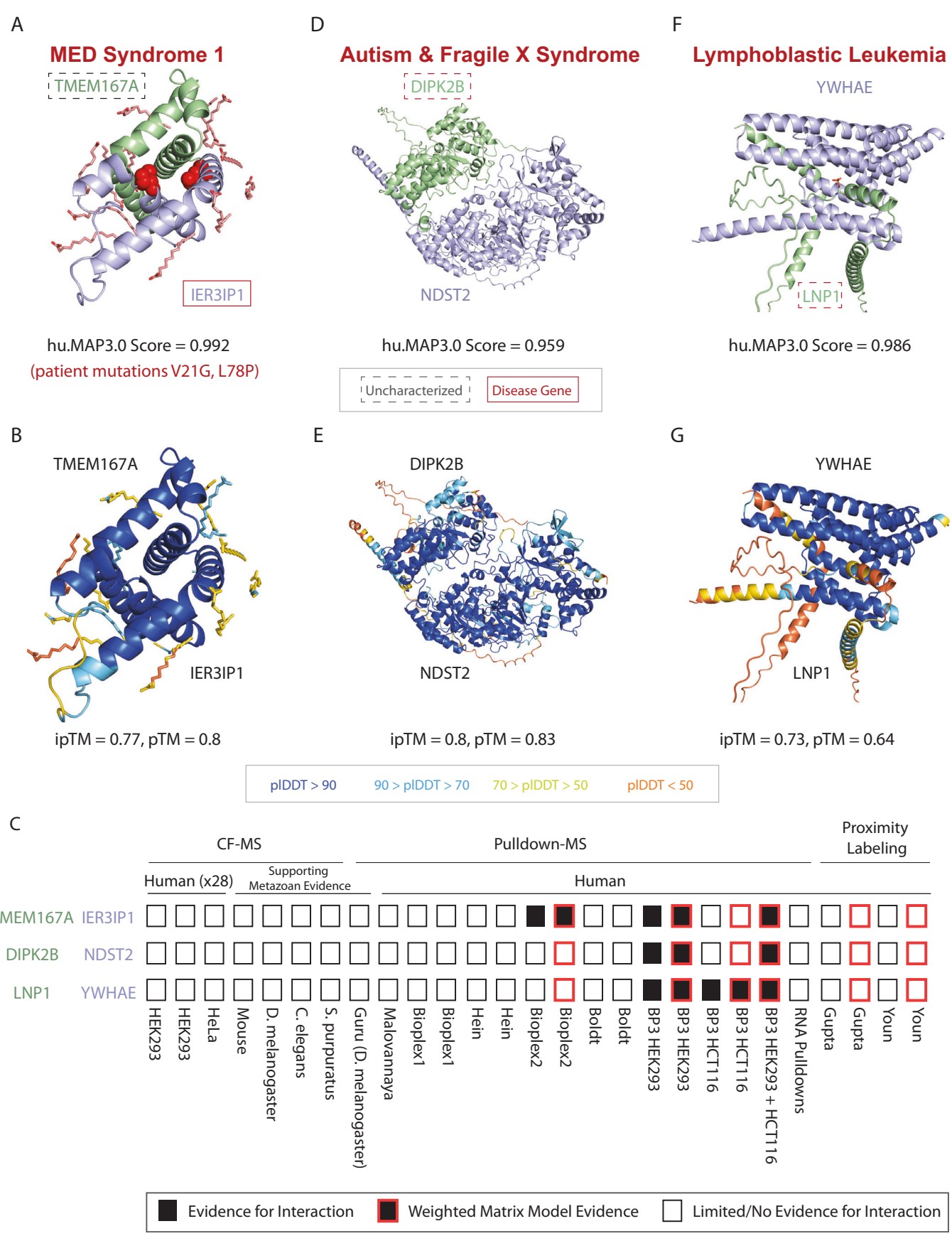

**A**

## MED Syndrome 1

TMEM167A

IER3IP1

hu.MAP3.0 Score = 0.992
(patient mutations V21G, L78P)

**D**

## Autism & Fragile X Syndrome

DIPK2B

NDST2

hu.MAP3.0 Score = 0.959

Uncharacterized   Disease Gene

**F**

## Lymphoblastic Leukemia

YWHAE

LNP1

hu.MAP3.0 Score = 0.986

**B**

TMEM167A

IER3IP1

ipTM = 0.77, pTM = 0.8

**E**

DIPK2B

NDST2

ipTM = 0.8, pTM = 0.83

**G**

YWHAE

LNP1

ipTM = 0.73, pTM = 0.64

pIDDT > 90   90 > pIDDT > 70   70 > pIDDT > 50   pIDDT < 50

**C**

| | | CF-MS | | | | | | | Pulldown-MS | | | | | | | | | | | | | | | | Proximity Labeling | | | |
|---|---|---|---|---|---|---|---|---|---|---|---|---|---|---|---|---|---|---|---|---|---|---|---|---|---|---|---|---|---|
| | | Human (x28) | | | Supporting Metazoan Evidence | | | | | | | | Human | | | | | | | | | | | | | | | |
| TMEM167A | IER3IP1 | ☐ | ☐ | ☐ | ☐ | ☐ | ☐ | ☐ | ☐ | ☐ | ☐ | ☐ | ☐ | ☐ | ■ | 🔴 | ☐ | ☐ | ■ | 🔴 | ☐ | ☐ | ☐🔴 | 🔴 | ☐ | | 🔴 | ☐ | 🔴 |
| DIPK2B | NDST2 | ☐ | ☐ | ☐ | ☐ | ☐ | ☐ | ☐ | ☐ | ☐ | ☐ | ☐ | ☐ | ☐ | ☐ | 🔴 | ☐ | ☐ | ■ | 🔴 | ☐ | ☐🔴 | 🔴 | ☐ | | 🔴 | ☐ | 🔴 |
| LNP1 | YWHAE | ☐ | ☐ | ☐ | ☐ | ☐ | ☐ | ☐ | ☐ | ☐ | ☐ | ☐ | ☐ | ☐ | ☐ | 🔴 | ☐ | ☐ | ■ | 🔴 | ■ | 🔴 | 🔴 | ☐ | | 🔴 | ☐ | 🔴 |
| | | HEK293 | HEK293 | HeLa | Mouse | D. melanogaster | C. elegans | S. purpuratus | Guru (D. melanogaster) | Malovannaya | Bioplex1 | Bioplex1 | Hein | Hein | Bioplex2 | Bioplex2 | Boldt | Boldt | BP3 HEK293 | BP3 HEK293 | BP3 HCT116 | BP3 HCT116 | BP3 HEK293 + HCT116 | RNA Pulldowns | | Gupta | Gupta | Youn | Youn |

■ Evidence for Interaction    🔴 Weighted Matrix Model Evidence    ☐ Limited/No Evidence for Interaction

◀ **Figure 5. Physical associations with uncharacterized proteins.**

(A) AlphaFold3 model of uncharacterized protein TMEM167A (green) and IER3IP1 (blue), a protein associated with Microcephaly Epilepsy Diabetes syndrome 1. Known patient mutations are shown in red spheres at the interaction interface. (B) Same model in (A), colored by pLDDT confidence score. (C) Graphical representation of evidence for protein interactions. (D) AlphaFold3 structural model of autism and Fragile X syndrome protein, DIPK2B (green), and NDST2 (blue) colored by chain. (E) Same model in (D), colored by pLDDT confidence score. (F) AlphaFold3 structural model of lymphoblastic leukemia protein, LNP1 (green), and YWHAE (blue). (G) Same model in (F), colored by pLDDT confidence score. Uncharacterized genes are highlighted with a dashed box. Disease genes are highlighted with a red box. Color scale for pLDDT is from blue (very high confidence) to orange (very low confidence).

protein TMEM167A (huMAP3_00596.1, hu.MAP3.0 score = 0.992) (Fig. 5A,B). IER3IP1 is associated with brain development and, when mutated, gives rise to developmental abnormalities including Microcephaly Epilepsy Diabetes syndrome (MEDS) (Abdel-Salam et al, 2012; Shalev et al, 2014; Poulton et al, 2011). At the cellular and molecular level, IER3IP1 localizes to the Golgi, regulates the endoplasmic reticulum (ER) and is involved in secretion of extracellular matrix proteins (Esk et al, 2020). We see evidence for this interaction from pulldown experiments as well as weighted matrix model analysis of pulldowns (Fig. 5C). Both IER3IP1 and TMEM167A are predicted to be transmembrane (Jones, 1999). Like IER3IP1, TMEM167A also localizes to the Golgi and is involved in protein secretion (Wendler et al, 2010). The ProteomeHD.2 network identifies a strong covariation between these two proteins (0.78) suggesting they are regulated as a module. Using AlphaFold3 (Abramson et al, 2024), we predicted a confident 3D model of IER3IP1 interacting with TMEM167A (ipTM = 0.77, pTM = 0.8) (Fig. 5A,B). Finally, we modeled two known missense mutations in IER3IP1 causative for MEDS in human patients (Val21Gly and Leu78Pro) (Poulton et al, 2011). Figure 5A shows these two mutations are at the interface of IER3IP1's interaction with TMEM167A suggesting a molecular mechanism for this disease.

Given this example, we next searched annotated hu.MAP3.0 complexes for the least characterized proteins to identify potential functions based on the "guilt-by-association" principle. Specifically, we searched for uncharacterized proteins (UniProt Annotation Score < 4) which were physically associated with complexes statistically enriched for function annotations (Fig. EV2E). Our analysis identifies potential functions for 472 of the least characterized human proteins (see Dataset EV5). Several of these functionally uncharacterized proteins are associated with human diseases including ciliary dyskinesia (CLXN (Hjeij et al, 2023)), Lui-Jee-Baron syndrome (SPIN4 (Lui et al, 2023)), lymphoblastic leukemia (LNP1 (Romana et al, 2006), SLX4IP (Meissner et al, 2014)), lethal skeletal dysplasia (TMEM263 (Mohajeri et al, 2021)), and association to autism and fragile X syndrome (DIPK2B (Aziz et al, 2011)).

The autism-associated protein, DIPK2B, is a member of the FAM69 family of protein kinases and has a signal peptide which localizes the protein to the extracellular region (Dudkiewicz et al, 2013). We identified DIPK2B as being co-complex with two proteins NDST2 and NDST3 (huMAP3_07022.1) involved in glycosaminoglycan biosynthesis (KEGG:00534). NDST2 and NDST3 are bifunctional enzymes which catalyze N-deacetylation and N-sulfation of glucosamine in the synthesis of heparin sulfate of the extracellular matrix (Aikawa and Esko, 1999; Duncan et al, 2006). Heparin sulfate deficiencies are thought to be causative for autism disease (Alexander et al, 2022). To provide further evidence for this interaction, we modeled the interaction between DIPK2B

and NDST2 (hu.MAP3.0 score = 0.959) using AlphaFold3 in which we obtained a confident structural model of the heterodimer (ipTM = 0.8) (Fig. 5D,E). Taken together, we propose that DIPK2B plays a role in maintaining heparin sulfate of the extracellular matrix and this is linked to its association with autism disorder.

We also identify LNP1, an uncharacterized protein, as associated with members of the 14-3-3 complex (huMAP3_06971.1). LNP1 is known to fuse with NUP98 in genome rearrangements of leukemia patients while the 14-3-3 complex is involved with many signaling pathways and associated with cancer (Fan et al, 2019). LNP1 has a known phosphoserine site at Ser114 (Ochoa et al, 2020) in a motif reminiscent of 14-3-3 binding (KFpSESF vs RXY/FXpSXP (Yaffe et al, 1997)). Provided this, we used AlphaFold3 to model the interaction between LNP1 and YWHAE, a 14-3-3 subunit which had the highest hu.MAP3.0 score 0.986 to LNP1. Figure 5F,G shows LNP1 and YWHAE confidently interact with a phosphorylated Ser114 interacting at the interface. This provides further evidence of LNP1's association with the 14-3-3 complex.

In total, our guilt-by-association transfer of function annotations results in a >70% increase from hu.MAP2.0 (Drew et al, 2021) (472 vs 274) in terms of the total number of uncharacterized proteins annotated. We find our analysis provides strong testable hypotheses for these understudied proteins and should guide follow-up studies.

## Identification of mutually exclusive subunits in hu.MAP3.0 complexes using AlphaFold structural models

Protein complexes often have modular components which, depending on the subunit present, alter the complex's function. When these multiple subunits exist, the subunits potentially compete for a binding interface and therefore cannot exist together in the same complex at the same time (i.e., mutually exclusive). Well known examples include replacement of PSMB5 for PSMB8 in the immunoproteasome (Abi Habib et al, 2022) (see above), ARID1A and ARID2 stabilizing the core of the chromatin remodeling PBAF complex, altering gene targeting (Hodges et al, 2016), and SINA and SINB in the Sin3 complex (Adams et al, 2020).

We next show multiple pieces of evidence can aid in the identification of mutually exclusive subunits. COPS7A and COPS7B are well supported as mutually exclusive with genetic and biochemical data as they differentially alter the function of the COP9 Signalosome (Adams et al, 2020; Zhou et al, 2021; Bech-Otschir et al, 2001). Using three-dimensional structures (Huang et al, 2016; Lingaraju et al, 2014) we observe COPS7A and COPS7B have overlapping interfaces with a third common subunit (Fig. EV4A,B). ProteomeHD.2 shows weak covariation between

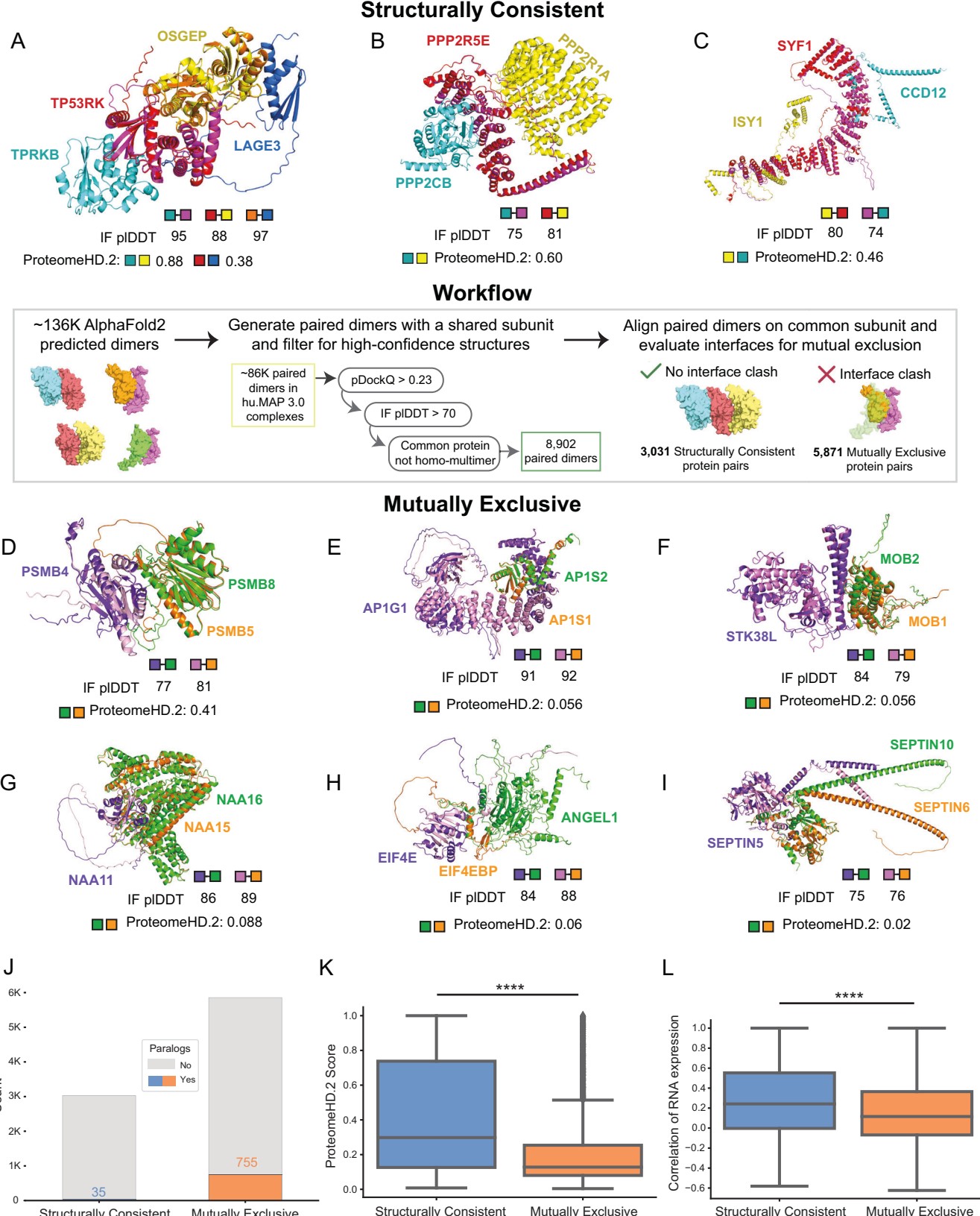

**Figure 6. Systematic structural analysis reveals mutually exclusive interactions.**

(Center panel) Overview of the workflow utilizing AlphaFold2 models to identify structurally consistent and mutually exclusive interactions within hu.MAP3.0 protein complexes. Structurally Consistent (**A–C**), Examples of interactions are shown with aligned AlphaFold2 models, where the common subunit (magenta and red) and the unique proteins (cyan and yellow) form structurally consistent interactions. The interfaces between the unique protein and common proteins do not overlap. IF pLDDT is displayed for each dimeric interface to indicate interface quality. The ProteomeHD.2 is displayed for the unique proteins to indicate their covariation across tissues. The Keops complex (**B**) has two structurally consistent trimers TPRKB-TP53RK-OSGEP and TP53RK-OSGEP-LAGE3, which is displayed as a tetramer. Mutually Exclusive (**D–I**), In contrast, mutually exclusive interactions are shown with aligned AlphaFold2 models, where the common protein subunit (light pink and purple) interacts with the unique proteins (green and orange) and the interfaces overlap. As before, interface quality metrics and the ProteomeHD.2 score for the unique protein subunits are shown. (**J**) Stacked barplot showing the prevalence of paralogs present in structurally consistent and mutually exclusive interactactions, with a higher frequency of paralogs found in mutually exclusive interactions. (**K**) Boxplot of the global distribution of ProteomeHD.2 scores between structurally consistent ($N = 2354$) and mutually exclusive ($N = 3733$) interactions, with structurally consistent interactions exhibiting stronger covariation compared to mutually exclusive interactions (Welch's $T$-test for independent samples, ****$P < 0.0001$, $P = 1.58 \times 10^{-151}$). Upper and lower bounds of the box correspond to quartiles of the distribution with first (Q1, 25th percentile) and third quartile (Q3, 75th percentile), respectively. The center line in the box corresponds to the median. The whiskers extend to the minimum and maximum values within $1.5 \times$ interquartile range (IQR) from Q1 and Q3, while potential outliers beyond 1.5× IQR are displayed as separate dots. (**L**) Boxplot comparing the correlation of RNA expression across different tissues for the unique proteins in structurally consistent ($N = 2980$) versus mutually exclusive ($N = 5801$) interactions (two-sided Mann-Whitney $U$-Test, ****$P < 0.0001$, $P = 2.22 \times 10^{-48}$). Upper and lower bounds of the box correspond to quartiles of the distribution with first (Q1, 25th percentile) and third quartile (Q3, 75th percentile), respectively. The center line in the box corresponds to the median. The whiskers extend to the minimum and maximum values within $1.5 \times$ interquartile range (IQR) from Q1 and Q3, while potential outliers beyond 1.5× IQR are displayed as separate dots.

COPS7A and COPS7B (ProteomeHD.2 score = 0.25) which is substantially lower than other COP9 subunit covariation signals (Fig. EV4C,D). This suggests COPS7A and COPS7B subunit expression patterns have altered to avoid conflicting functions and may be an indicator of mutual exclusion.

We next asked if AlphaFold models were of sufficient confidence to identify mutually exclusive proteins. ABRAXAS1 and ABRAXAS2 are members of the BRISC/BRCA1-A complex (Fig. EV4E). A structural model of the complex in Fig. EV4F shows ABRAXAS1 and ABRAXAS2 overlap at a shared interface suggesting they are mutually exclusive. Further we observe that ABRAXAS1 and ABRAXAS2 have limited covariance (ProteomeHD.2 score = 0.096), consistent with the idea that the two proteins avoid competing for the same binding interface using different protein expression patterns. We therefore can place ABRAXAS2 in the BRISC complex (Fig. EV4G) and ABRAXAS1 in the BRCA1-A complex (Fig. EV4H). Additionally, we observe an uncharacterized protein, C9orf85, as being physically associated with core subunits of the BRISC/BRCA1-A complex. We see strong evidence for a physical interaction between C9orf85 and ABRAXAS2 (hu.MAP3.0 score = 0.989) but limited evidence for C9orf85 being physically associated with ABRAXAS1 (hu.MAP3.0 score = 0.0). This points to C9orf85 as a member of the BRISC complex and potentially contributes to its specific function (Fig. EV4G).

As shown in these examples, it is highly beneficial to have three-dimensional structures of protein complexes to evaluate mutually exclusive subunits. Several studies have applied AlphaFold2/AlphaFold-multimer workflows to determine structures of thousands of protein interactions (Trip et al, 2025; Burke et al, 2023; Jänes et al, 2024). We next utilize these compendiums of pairwise structural models to systematically identify potential mutually exclusive pairs in hu.MAP3.0 complexes (Fig. 6, center panel). We first filtered the structural models to determine confident predictions. We then identified confident models that share a common subunit and aligned on the common subunit. Finally, we evaluated the interfaces of proteins with the common subunit for potential overlap. If an overlap of proteins exists, the proteins are labeled mutually exclusive. Alternatively, if no overlap exists at the interface the interaction is labeled structurally consistent (see

Methods). Using this method we identify 5871 mutually exclusive and 3031 structurally consistent pairs of proteins in hu.MAP3.0 complexes (Dataset EV6).

Figure 6A–C shows examples of structurally consistent interactions identified by our workflow. This highlights unique interfaces for pairs of proteins sharing a common third protein. These models include the PP2A complex with a structural model for the catalytic subunit PPP2CB and interactions from the KEOPS complex, which has been previously modeled in yeast (Humphreys et al, 2021) and human using homology modeling (Drew et al, 2017b). Further, we identify members of the spliceosomal complex with an uncharacterized protein, CCDC12. We observe high pairwise ProteomeHD.2 scores of structurally consistent interactions suggesting they covary as protein modules.

Figure 6D–I shows examples of proteins we labeled as mutually exclusive interactions. We identify positive control examples PSMB5 and PSMB8 known to substitute for each other in the proteasome (Inada and Beckmann, 2024) (Fig. 6D). We also identify other pairs that are supported by the literature to be mutually exclusive. AP1S1 and AP1S2 have distinct, non-redundant functions that lead to different disease pathologies when mutated (Montpetit et al, 2008; Tarpey et al, 2006) (Fig. 6E). MOB1A and MOB2 have previously been shown to compete for binding on STK38 (Kulaberoglu et al, 2017), a paralog of STK38L (Fig. 6F). NAA16 has been suggested to substitute for NAA15 and carry out similar functions in NAA15-knockdown experiments (Arnesen et al, 2005) (Fig. 6G).

Biochemical data shows ANGEL1 as an interaction partner of EIF4E, and uses a similar binding motif as EIF4EBP1 (Yanagiya et al, 2012) (Fig. 6H). Despite their mutual exclusivity, overexpression of either protein does not reduce binding of the other. This is likely due to their distinct cellular localizations where EIF4EBP1 localizes to the cytoplasm and ANGEL1 is confined to the golgi apparatus and endoplasmic reticulum (Yanagiya et al, 2012). This suggests regulation of subcellular localization plays a key role in maintaining this exclusivity.

We also identify interactions of septin complexes, which form various filament-forming, hetero-oligomeric complexes and are crucial components of the cytoskeleton, such as protein scaffolding and diffusion barriers (Mostowy and Cossart, 2012). The SEPTIN2-

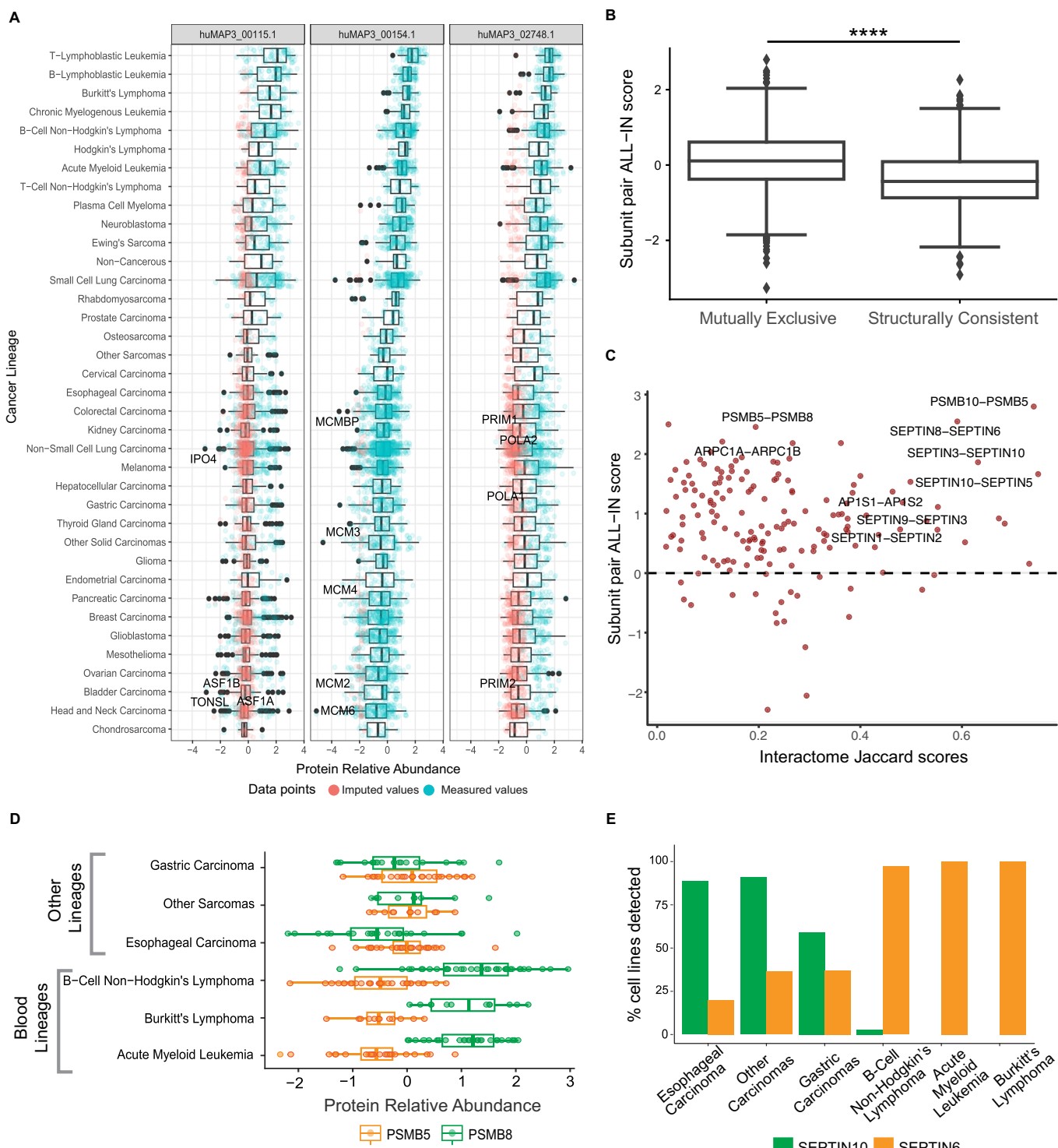

**Figure legend**

© The Author(s)

*Molecular Systems Biology* Volume 21 | Issue 7 | July 2025 | 911–943    **923**

SEPTIN6-SEPTIN7 complex is the most studied and abundant septin assembly (Wang et al, 2018). However, based on sequence homology, previous research suggests that subunits within this complex are often substituted (Kinoshita, 2003). For example, SEPTIN5, SEPTIN4, or SEPTIN1 can replace SEPTIN2, while SEPTIN8, SEPTIN10, or SEPTIN11 can replace SEPTIN6. Pull down experiments further suggest interactions of SEPTIN5 with

SEPTIN6 and SEPTIN10 (Huttlin et al, 2015), though many other septins are detected as well, likely reflecting their filamentous assembly. Our findings uniquely reveal mutual exclusivity between SEPTIN 10 and SEPTIN6 when binding to SEPTIN5 suggesting they compete for a shared interface (Fig. 6I).

We next asked if proteins involved in mutually exclusive interactions are enriched with Gene Ontology annotations. This

**Figure 7. Lineage-specific expression of mutually exclusive subunits.**

(A) Relative protein expression across cancer lineages for three hu.MAP3.0 complexes ($N = 10{,}439$, $N = 4745$, $N = 5694$, in order as plotted from left to right) that exhibit a high expression in leukemia lineages. Each point represents one subunit in one cell line. Where a subunit was not detected in a particular cell line, a low abundance value has been imputed. Boxplots show the distribution of relative expression. Upper and lower bounds of each box correspond to quartiles of the distribution with first (Q1, 25th percentile) and third quartile (Q3, 75th percentile), respectively. The center line in the box corresponds to the median. The whiskers extend to the minimum and maximum values within $1.5 \times$ interquartile range (IQR) from Q1 and Q3, while potential outliers beyond 1.5× IQR are displayed as separate black dots. (B) Boxplot of the expression-based mutual exclusivity score (ALL-IN) for mutually exclusive ($N = 995$) interactions and structurally consistent ($N = 1170$), respectively. Mutually exclusive subunits have a significantly higher ALL-IN score (Welch's $T$-test for independent samples, ****$P < 0.0001$, $P = 8.87 \times 10^{-44}$). Upper and lower bounds of the box correspond to quartiles of the distribution with first (Q1, 25th percentile) and third quartile (Q3, 75th percentile), respectively. The center line in the box corresponds to the median. The whiskers extend to the minimum and maximum values within $1.5 \times$ interquartile range (IQR) from Q1 and Q3, while potential outliers beyond 1.5× IQR are displayed as separate dots. (C) Scatter plot of a subset of protein pairs from hu.MAP3.0 complexes, separated by their hu.MAP3.0 pairwise interactome overlap along the x-axis, and by their expression-based mutual exclusivity score (ALL-IN) along the y-axis. For pairs in the top right, a high ALL-IN score provides strong support for structure-predicted mutual exclusivity, and the pairs share many interaction partners. For visualization purposes, for each protein, the best scoring subunit pair in ALL-IN score is shown, with all pairs available in the supplementary table. (D) Boxplots of relative protein abundance levels across the top 3 lineages that exhibit low and high expression of PSMB5 ($N = 132$) and PSMB8 ($N = 119$), respectively. Upper and lower bounds of the box correspond to quartiles of the distribution with first (Q1, 25th percentile) and third quartile (Q3, 75th percentile), respectively. The center line in the box corresponds to the median. The whiskers extend to the minimum and maximum values within $1.5 \times$ interquartile range (IQR) from Q1 and Q3. Individual data points are plotted. (E) Percentage of cell lines per cancer lineage that express the SEPTIN10 and SEPTIN6 proteins at detectable levels. Number of cell lines per lineage are shown above the bars.

may point to certain biological systems whose functions are more readily regulated by substitutions of subunits. To determine potential functional enrichment signatures of mutual exclusivity, we analyzed Gene Ontology annotations of all proteins identified in our mutually exclusive dataset (Fig. EV5A). We found enrichment in terms such as "intracellular vesicle", "cell junction", and "signal transduction". These align with our highlighted examples above, AP1 complex (vesicle transport), septin interactions (cellular junction) and STK38L-MOB1A-MOB2 interactions (signaling). Overall this suggests these biological systems reuse core components and alter their function by substitution of modular subunits.

Studies of individual complexes (Adams et al, 2020) as well as systematic investigations in yeast have suggested protein paralogs as mutually exclusive (Szklarczyk et al, 2008). We therefore asked how prevalent paralogs are in structurally consistent and mutually exclusive protein pairs. Figure 6J shows a substantial enrichment of paralogs in the mutually exclusive set where 755 out of 5871 pairs (~13%) are paralogs while only 35 out of 3031 (~1%) structurally consistent are paralogs. This suggests gene duplication is a common way for complexes to alter their functions through the subfunctionalization of paralogs.

We next investigate how cells have adapted to the scenario of multiple subunits competing for the same interface. We show in the examples above, mutually exclusive subunits have altered expression patterns as well as altered subcellular localization. To determine the degree to which expression plays a role, we compared ProteomeHD.2 scores of mutually exclusive protein pairs to protein pairs that are structurally consistent. Figure 6K shows mutually exclusive pairs are more likely to have lower ProteomeHD.2 covariation values than structurally consistent proteins. We additionally observe lower RNA expression correlation (Fig. 6L) and higher tissue specificity (Fig. EV5B) among mutually exclusive subunits relative to structurally consistent pairs. To explore subcellular localization as an alternative regulator of mutual exclusivity, we compared subcellular localization annotations from HPA (Uhlén et al, 2015) of mutually exclusive pairs as well as annotations of structurally consistent pairs (Fig. EV5C). Although the difference between the two distributions is statistically significant ($p < 0.0001$), the overall shape and patterns of the distributions are largely similar, suggesting that changes in

subcellular localization are not as prominent as changes in expression. Taken together, this suggests that mutually exclusive subunits are regulated primarily at the level of expression followed by changes in subcellular localization.

## Lineage-specific expression of protein complexes and mutually exclusive subunits

Given the critical role of expression regulation in shaping complexes with specialized functions through mutually exclusive subunits, we next explored the potential implications of context-specific expression of these subunits in cancer lineages. To investigate this, we turned to a recent large-scale study that reported proteomes for 949 human cancer cell lines, derived from 28 tissues and over 40 different cancer types (Gonçalves et al, 2022). We first examined whether hu.MAP3.0 complexes exhibited differential expression between cancer lineages and identified complexes whose expression most varies across the different cancer lineages (Appendix Fig. S2). Among them, the MCM complex (huMAP3_00154.1), which is more abundant in blood cancer lineages, including leukemias and lymphomas (Fig. 7A; Appendix Fig. S3). Although from this dataset alone we cannot determine whether higher abundance of MCM reflects disease progression, upregulation of the MCM complex has previously been associated with poor overall survival in lymphoma (Marnerides et al, 2011).

While our analysis revealed that cancer lineages vary significantly in terms of how many complexes are differentially expressed (Appendix Fig. S3), most hu.MAP3.0 complexes do not show lineage-specific expression variation as a unit (Appendix Fig. S2). We therefore assessed whether individual subunits of these complexes may be expressed in a lineage-specific manner potentially altering complex function. We therefore focussed on our identified mutually exclusive subunits. For all subunits of hu.MAP3.0 complexes we calculated a pairwise mutual exclusivity score, which we named ALL-IN (Average celL Lineage co-expressIoN) score, as it integrates the following four expression-based features (see Methods): (1) correlation of protein abundances across the 949 cancer cell lines; (2) correlation of protein abundances aggregated by lineage; (3) a metric capturing very large expression differences (expression vs non-expression); (4)

**Table 1. Updates from hu.MAP2.0.**

|  | hu.MAP2.0 | hu.MAP3.0 |
|---|---|---|
| Model | SVM Classifier | Auto-ML (AutoGluon) |
| Data (experiments) | 15,806 | 25,575 |
| Benchmark | Corum | Complex Portal |
| Features | 292 | 324 |
| Proteome coverage | 9963 | 13,769 |
| Number of complexes | 6965 | 15,326 |
| Unique interactions | 57,148 | 159,451 |

correlation with other subunits of the complex. When evaluating the ALL-IN score of our mutually exclusive and structurally consistent pairs, we observed a distinct and statistically significant separation between the two groups, indicating that mutually subunits are more likely to exhibit distinct expression profiles across cancer lineages, serving as a potential indicator of mutual exclusivity (Fig. 7B). To evaluate potential error due to spurious protein pairs, we calculate the Jaccard Interactome score, which assesses how likely pairs of proteins have the same complex members (see Methods). Pairs with high ALL-IN scores and Jaccard Interactome scores are most likely to be true mutually exclusive interactions (Fig. 7C). For these protein pairs, we thus provide multiple lines of evidence to support their function as mutually exclusive complex subunits: shared hu.MAP3.0 complex membership, overlapping structural models, and divergent expression in cancer lineages. Furthermore, for pairs that are covered by the cancer cell lineage dataset, it may be possible to link context-specific versions of protein complexes to different cancer lineages. For example, the PMSB8 subunit of the immunoproteasome is more abundant in blood lineages than PSMB5, the canonical proteasome subunit it replaces in these cell types (Fig. 7D). Similarly, we identified SEPTIN6 and SEPTIN10 as mutually exclusive interactors of SEPTIN5 (Figs. 6I and 7C). It has been previously noted that Septins are distributed differently across cell types, which could suggest expression as a potential regulator of these mutually exclusive interactions (Mostowy and Cossart, 2012). Indeed, we observe that SEPTIN6 is expressed in blood lineages, but is not detectable in lineages where SEPTIN10 is detected, such as gastric and esophageal lineages (Fig. 7E). Importantly we demonstrate how the ALL-IN score can capture both protein pairs that anti-correlate in their expression patterns (Fig. 7D) but also those that are not expressed in specific lineages highlighting their mutual exclusivity (Fig. 7E).

## Discussion

Here we describe our integration of >25,000 proteomic experiments to build a more complete and accurate set of protein complexes. We advance on our previous work in several aspects (Table 1). First, we place 13,769 human proteins into at least one protein complex. This is an increase of ~40% over hu.MAP2.0. Second, we evaluate our protein complexes with respect to the protein covariation network, ProteomeHD.2. Protein covariation networks demonstrate considerable power in the identification of protein functional modules. Moreover, the ProteomeHD.2 database is compiled from

datasets independent from those used to construct hu.MAP3.0. Therefore, interactions that agree between the two networks are more confident as there are multiple independent lines of evidence for their existence. Third, we developed a workflow using structural modeling to identify mutually exclusive pairs of proteins that likely do not simultaneously participate in a complex. This set of protein pairs point to potential complexes with multiple functions. As structural models of protein interactions become more widespread, we anticipate to find many more mutually exclusive pairs in protein complexes. Finally, we develop the ALL-IN score which is a prediction score of mutual exclusivity for protein pairs based on protein interactions and expression data. This is a general approach that can be applied to new interaction networks and expression datasets as they become available.

It is important to note the limitations of our complex map. For example, the coverage of hu.MAP3.0 does not yet cover the complete human proteome. Moreover, hu.MAP3.0 is constructed from experiments from many different cell types and tissues, thereby limiting our resolution of cell type specific interactions. We anticipate the coverage of future protein complex maps to improve as experimental proteomics datasets sample more of the proteome. Further, we expect the hu.MAP3.0 resource to aid development of cell type specific complex maps by utilizing our overall atlas with context-specific experiments.

To explore caveats of our model further we looked at false positives and false negatives predicted by our model determined by our test set. An illustrative example of a false positive is CPSF6 and CPSF7, which are known components of separate variants of the cleavage factor I (CFIm) 3′ end processing complex. Our model predicts CSPF6 and CSPF7 to interact with a confidence score = 0.99. This prediction is driven by reciprocal pulldowns of CSPF6 and CSPF7 in Bioplex and Hein et al (among other weighted matrix model evidence). There are other examples of our model predicting high confidence interactions among negative interactions which are variants of known complexes. Examples include the SWI/SNF complex components BCL7C-SMARCC2 (0.99), the Adaptor-1 complex components AP1S1-AP1S2 (0.96), and the NatA complex components NAA15-NAA16 (0.99). We show AP1S1-AP1S2 and NAA15-NAA16 are mutually exclusive based on structural modeling and therefore unlikely to be present in the same complex (Fig. 6E,G). We also observe high confident negative interactions are more likely to share gene ontology terms than random (Appendix Fig. S4). We therefore see two possibilities arise to explain high confident negative interactions. First, our model has difficulty distinguishing complex variants, many of which share gene ontology terms. Our mutually exclusive analysis aids in identifying these variants. Second, high confident negative interactions are actually true interacting pairs that have yet to be annotated as in the same complex. We suggest these high confident negative interactions to be prioritized in future annotation efforts.

With respect to false negative examples, we observe our model generates low confidence scores for several test positive interactions largely due to lack of data supporting the interactions. Interestingly, some of these known interactions such as PARD3-PARD6A (confidence score = 0.06) and ENY2-TAF4 (confidence score = 0.003) are identified in the same hu.MAP3.0 complex. Both PARD3 and PARD6A have high confidence interactions with PRKCI in complex huMAP3_05806.1. ENY2 and TAF4 have high confidence interactions with TAF10 in complex huMAP3_09451.1. In fact

~10% of low confidence positive interactions are placed in the same complex based on confident interactions with other partners. This suggests our clustering approach mitigates the lack of data for certain interactions increasing the precision for low confidence interactions. In fact, after clustering, a hu.MAP3.0 confidence score of 0.1 increases precision to >0.6 (Fig. EV2D). We therefore include low confidence pairs that have clustered together in our final network with their confidence score (Dataset EV7).

Mutually exclusive interactions introduce a conflict for organisms to resolve as two proteins are in competition for a single interface. This has been previously investigated by Kim et al, who mapped yeast interactomes onto homologous known structures from the PDB (Kim et al, 2006). Here, we build on this idea by using structural modeling to identify mutually exclusive interactions and investigate how these structural constraints are managed within the cell. We observed the dominant approach for organisms to relieve this conflict is to alter the expression of mutually exclusive pairs so they are not present concurrently in the same cell (Fig. 6K,L). Other possible mechanisms for organisms to relieve this conflict include altering the subcellular localizations of mutually exclusive pairs and regulating protein subunits at the level of post translational modifications (PTMs). We see limited evidence for mutually exclusive pairs having altered their subcellular localization compared to structurally consistent pairs (Fig. EV5C). This is in line with other groups who have explored whether paralogs in yeast have differing subcellular localization and found duplicated genes are not more likely to relocalize compared to singleton genes (Qian and Zhang, 2009). Post translational modification of individual mutually exclusive subunits is another potential mechanism to activate or deactivate a protein subunit's membership in a complex. Many interactions are known to be modulated by PTMs (Csizmok and Forman-Kay, 2018). Although we don't expect PTMs to play as large of a role as expression, we do anticipate exploring this mechanism further in the future.

Another possibility to relieve the conflict of mutually exclusive pairs is protein multimerization. Proteins may relieve the conflict by having multiple binding sites (e.g., multimerize the common protein) to accommodate the two mutually exclusive proteins. For example, proteins that form fibers such as actin can support many proteins binding to the same interface yet at different protein instances along the fiber. Here we partially mitigate this issue by filtering out mutually exclusive interactions where the common protein likely homo-multimerizes. We do not rule out examples where the common protein forms fibers by hetero-multimerization. We feel these are interesting cases of mutually exclusive / structurally consistent interactions worthy of understanding in the future.

Jänes et al recently studied mutually exclusive interactions of proteins with 5 or more interaction partners. They discuss this analysis may have a higher rate for paralogs due to paralogs being included in the AlphaFold-multimer multiple sequence alignments (Jänes et al, 2024; Burke et al, 2023). We observe an enrichment in paralogs in our set of mutually exclusive interactions (~13%) and acknowledge the potential for error when evaluating paralogs. We do, however, note there are well characterized paralogs with experimental structures that are known to be mutually exclusive (e.g., PSMB5-PSMB8, COPS7A-COPS7B, ABRAXAS1-ABRAXAS2). This suggests that paralogs are still likely major participants of mutually exclusive interactions. Ultimately, new

methods will be required to refine structurally modeling of paralogs.

Also implicit in our definition of mutual exclusive pairs is that the pairs are required to sterically hinder each other. An alternative class of mutually exclusive pairs may include those that allosterically alter the common protein surface to restrict the binding of the other protein. Additional modeling of alternative conformations is likely required to identify this alternative class. We also make sensible cut-offs such as quality of AlphaFold2 models (i.e., IF pLDDT, pDockQ) and number of overlapping interface residues to identify mutually exclusive pairs. We therefore anticipate as structural modeling of protein interactions becomes more accurate, we will identify additional mutually exclusive interactions.

Finally, we imported the 15,326 hu.MAP3.0 clusters into the Complex Portal and matched at the canonical level to existing manually curated entries. No thresholds were imposed in the minimum number of participants required to make the match. Where the protein components are an exact match to an existing manually curated complex, the entries have been merged and the hu.MAP3.0 accession number retained as a cross-reference in the entry, with the qualifier 'identity'. Where the hu.MAP complex is a subset of an existing manually curated entry, or entries, again these have been merged and the hu.MAP3.0 cross-reference added, with the qualifier 'subset'. The remaining complexes were created as novel assemblies into the Complex Portal (Balu et al, 2025) and have been publicly released for search and download. The complexes will be priority targets for manual curation based on literature curation and comparison to small-scale interaction datasets in the IMEx Consortium dataset (Porras et al, 2020), and to other curated datasets in resources such as PDB (wwPDB consortium, 2019), EMDB (wwPDB Consortium, 2024) and Reactome (Milacic et al, 2024). The use of LLMs to enable rapid curation of these assemblies based on the presence of sets of gene names in a publication is also being evaluated. Overall, we anticipate this to be an extremely valuable tool for the broad research community.

## Methods

**Reagents and tools table**

| Reagent/Resource | Reference or Source | Identifier or Catalog Number |
| --- | --- | --- |
| **Experimental models** | | |
| N/A | | |
| **Recombinant DNA** | | |
| N/A | | |
| **Antibodies** | | |
| N/A | | |
| **Oligonucleotides and other sequence-based reagents** | | |
| N/A | | |
| **Chemicals, Enzymes and other reagents** | | |
| N/A | | |
| **Software** | | |
| MCL Version: 14-137 | https://micans.org/mcl/ (Enright et al, 2002) | |

| Reagent/Resource | Reference or Source | Identifier or Catalog Number |
|---|---|---|
| ClusterOne Version: 1.0 | https://paccanarolab.org/clusterone/ (Nepusz and Paccanaro, 2012) | |
| ProteinComplexMaps Version: bdea06b | https://github.com/KDrewLab/ protein_complex_maps (Drew et al, 2021) | |
| AutoGluon Version 0.4.0 | https://auto.gluon.ai/stable/install.html (Erickson et al, 2020) | |
| Training of hu.MAP3.0 with AutoGluon | https://github.com/KDrewLab/ huMAP3.0_analysis | |
| hu.MAP3.0 Machine learning model | https://huggingface.co/DrewLab/ hu.MAP_3.0_AutoGluon | |
| Train and test feature matrices | https://huggingface.co/datasets/DrewLab/ hu.MAP_3.0 | |
| EvalMutualExclusivity | https://github.com/KDrewLab/ huMAP3.0_analysis | |
| Cancer lineage expression analysis in hu.MAP3.0 complexes | https://github.com/kustatscher-lab/ HuMAP3_ProHD2 | |
| **Other** | | |
| AlphaFold2 structures for mutually exclusive interaction modeling | https://archive.bioinfo.se/huintaf2/humap.zip (Burke et al, 2023) https://ftp.ebi.ac.uk/pub/databases/ProtVar/ predictions/interfaces/ 2024.05.28_interface_models_high_confidence.tar (Jänes et al, 2024) | |

## Curation of mass spectrometry datasets

To build the hu.MAP3.0 network, we integrated data from more than 25,000 previously published mass spectrometry experiments. Leveraging our specialized machine learning framework, we enhanced our previously published hu.MAP2.0 network by incorporating ~10,000 new affinity purification data from Bioplex 3.0 (Huttlin et al, 2021). Bioplex 3.0 expands ORFeome coverage with over 10,000 baits in HEK293T cells and biological diversity by including HCT116 cells, significantly enriching the interactome within our network.

To construct a machine learning classifier, we collected mass spectrometry data from multiple publications and derived features predictive of protein interactions. Protein interaction features for datasets used in hu.MAP2.0 (Drew et al, 2021) were downloaded

from https://zenodo.org/records/15293715/files/humap2_feature_matrix_20200820.featmat.gz?download=1. This matrix, containing 292 features, has been previously described (Drew et al, 2021). Briefly, the features were obtained from original sources or calculated directly from raw mass spectrometry data. Of these, 220 features are derived from 55 CF-MS datasets (Wan et al, 2015) and include metrics such as Poisson noise Pearson correlation coefficient, weighted cross-correlation, co-apex score, and MS1 ion intensity distance. Twenty-nine features are sourced from AP-MS data, including HGSscore (Guruharsha et al, 2011), MEMOs (core modules) certainty assignments (Malovannaya et al, 2011), and metrics from Bioplex 1.0 (Huttlin et al, 2015) and 2.0 (Huttlin et al, 2017), such as NWD Score, Z Score, Plate Z Score, Entropy, Unique Peptide Bins, Ratio, Total PSMs, Ratio Total PSMs, and Unique:Total Peptide Ratio. Additional AP-MS features include prey.bait.correlation, log10.prey.bait.ratio, log10.prey.bait.expression.ratio, mean psm (Hein et al, 2015), and socioaffinity index terms (Boldt et al, 2016). Fifteen features come from proximity labeling datasets (Youn et al, 2019; Gupta et al, 2015), including metrics Average Spectra, Average Saint probability, Max Saint probability, Fold Change, and Bayesian FDR estimate. Data sources for datasets incorporated into hu.MAP3.0 are highlighted in Table 2.

Newly integrated Bioplex 3.0 datasets comprised the same measures that were used for Bioplex 2.0 and Bioplex 1.0 features obtained (see Table 2), including specifically NWD Score, Z Score, Plate Z Score, Entropy, Unique Peptide Bins, Ratio, Total PSMs, Ratio Total PSMs, Unique:Total Peptide Ratio and Average Assembled Peptide Spectral Matches. The features are obtained from both the HEK293T and HCT-116 datasets.

Additional features were generated using a weighted matrix model (WMM), which is based on the hypergeometric distribution as described in (Hart et al, 2007), and previously used in (Drew et al, 2021, 2017a). WMM features are derived from the presence or absence of proteins across individual experiments. To mitigate the effects of noise common in high-throughput mass spectrometry data, such as the spurious identification of proteins with low counts in single experiments, we applied thresholds to ensure that only high-quality identifications were considered. The datasets utilized were filtered according to the protocols established in (Drew et al, 2021). We calculated WMM features for each BioPlex 3.0 cell type

**Table 2.  MS Datasets incorporated into hu.MAP3.0.**

| Dataset | Number of Experiments | Link |
|---|---|---|
| Wan et al (2015) | 5344 fractions | http://hu1.proteincomplexes.org/static/downloads/feature_matrix.txt.gz |
| Hein et al (2015) | 1125 bait pull-downs | http://hu1.proteincomplexes.org/static/downloads/feature_matrix.txt.gz |
| Gupta et al (2015) | 2 conditions × 58 proximity labeled baits = 116 | http://prohits-web.lunenfeld.ca/ |
| Youn et al (2018) | 119 proximity labeled baits | http://www.cell.com/cms/attachment/2118963855/2087347233/mmc2.xlsx |
| Boldt et al (2016) | 217 bait pull-downs | https://static-content.springer.com/esm/art%3A10.1038%2Fncomms11491/MediaObjects/ 41467_2016_BFncomms11491_MOESM835_ESM.xlsx |
| Treiber et al (2017) (reprocessed by (Mallam et al, 2019)) | 3004 RNA hairpin pull-down MS runs | https://www.cell.com/cms/10.1016/j.celrep.2019.09.060/attachment/45abb95b-ef3f-4752-8906-dc5eed118480/mmc4.csv |
| Bioplex 3.0 (Huttlin et al 2021) (includes Bioplex 2.0 and 1.0) | 10,128 bait pull-down HEK 293T, 5522 bait pull-downs HCT 116 | https://bioplex.hms.harvard.edu/data/ BioPlex_BaitPreyPairs_noFilters_293T_10K_Dec_2019.tsv, https://bioplex.hms.harvard.edu/ data/BioPlex_BaitPreyPairs_noFilters_HCT116_5.5K_Dec_2019.tsv |

as well as a combined set (i.e., HEK293T, HCT-116, HEK293T/HCT-116). Additionally, we calculated WMM features for two abundance thresholds, where a given protein was required to have >2.0 Bioplex3.0 $Z$ score and >4.0 Bioplex3.0 $Z$ score. For each calculation, we generate a feature in the form of the negative natural log $p$-value of Eq. (1) and the total number of experiments the pair of proteins is observed together (i.e., pair count). The final feature matrix contains 324 features for 25,992,008 protein pairs.

$$p(\#shared\ experiments \geq k|n,m,N) = \sum_{i=k}^{min(n,m)} \frac{\binom{n}{i}\binom{N-n}{m-i}}{\binom{N}{m}}$$

(1)

## Gold standard protein complex test and train sets

To generate a test and training set of protein complexes, we downloaded the complete set of human protein complexes from the Complex Portal (Meldal et al, 2019), version 2022_07_11 (https://ftp.ebi.ac.uk/pub/databases/intact/complex/2022-07-11/). The Complex Portal is a manually curated database of stable protein complexes. Isoforms were removed. Complexes were randomly assigned to the test or train set. To prevent overlap between these sets, any overlapping complexes (i.e., those sharing protein pairs) were identified and the largest complex was removed, ensuring that no PPI was present in both sets. Additionally, complexes greater than 30 subunits were removed to prevent large protein complexes from dominating the test and training sets. Within the test and train sets, pairs of proteins were generated and labeled as positive and negative. Positive pairs are proteins within the same complex; negative pairs are proteins in separate complexes. Any intersecting PPIs between the test and training sets were identified and removed to prevent overlap. The split_complexes.py script in the python package protein_complex_maps (Reagents and Tools Table) was used for test and training set generation. Below are annotated command-line examples used to reduce redundancy in the gold standard complex lists and split complexes into test and training sets:

### Reduce redundancy of benchmark complexes

To remove redundancy and remove bias within benchmark complexes, we first remove all complexes larger than 30 subunits including any subcomplexes that may exist. This eliminates several large complexes such as the ribosome and proteasome which otherwise dominate the benchmark. We then iteratively compare all complexes to all other complexes calculating pairwise Jaccard coefficients. For any pair of complexes with a Jaccard coefficient above a threshold (0.6) we remove the larger complex of the pair. The process is repeated until no pair of complexes' Jaccard coefficient is above the threshold. In practice, the benchmark complex file was formatted as tab separated lists of UniProt identifiers, with one complex per line. We then ran the python script complex_merge.py using the –remove_largest flag which removes the largest of redundant pairs of complexes. The input file formatted from Complex Portal can be found in Dataset EV8.

```
python ./protein_complex_maps/preprocessing_util/
complexes/complex_merge.py --cluster_filename
```

```
ComplexPortal_human_complexes.txt --output_filename
ComplexPortal_human_complexes_reduced.txt --merge_
threshold 0.6 --complex_size_threshold 30 --remove_
large_subcomplexes --remove_largest
```

### Generating test and train sets

Once benchmark complexes are removed of redundancy, we generate test and training sets. Specifically, the list of complexes is randomly shuffled and split into test and training sets, 50:50. We generate positive and negative pairs from the test and training sets. A positive pair is defined as two proteins present in the same complex. A negative pair is defined as two proteins present in separate complexes and never in the same complex in either test or training. We eliminate any redundancy between test and training positive pairs and test and training negative pairs by randomly removing redundant pairs. We finally compare all complexes in the training set to all complexes in the test set identifying complexes that overlap by two proteins (protein pair). If an overlap is detected, we randomly remove an overlapping complex from either the test or training sets. In practice the complex file is used as input into the split_complexes.py script, which generates six output files: negative training PPIs (input_filename.neg_train_ppis.txt), negative test PPIs (input_filename.neg_test_ppis.txt), positive training PPIs (input_filename.train_ppis.txt), positive test PPIs (input_filename.test_ppis.txt), the complexes used for training PPI generation (input_filename.train.txt), and the complexes used for test PPI generation (input_filename.test.txt). All six resultant files used in this analysis can be found in Dataset EV8.

```
python ./protein_complex_maps/preprocessing_util/
complexes/split_complexes.py --input_complexes
ComplexPortal_human_complexes_reduced.txt --random_
seed 1234 --size_threshold 30 --remove_large_
complexes --remove_largest --merge_threshold 0.6
```

## Model description

To generate a protein interaction network, we utilized AutoGluon (Erickson et al, 2020), an open-source automated machine learning framework, to classify pairs of proteins as co-complex interactions. AutoGluon efficiently explores a broad range of model architectures, selecting those that best optimize the predictive accuracy of the final model.

To train the model, we constructed a feature matrix with protein pairs as rows and features as columns. Positive and negative labels were assigned using the benchmark training set. We randomly subsampled 10,000 negative labeled pairs to balance positive and negative labels. We used the AutoGluon TabularPredictor module using the constructed feature matrix to fit a model that discriminates between positive and negative interactions. We evaluated three different evaluation metrics, including accuracy, F1, and precision. The final model was selected based on its AUPRC (area under the precision-recall curve) on the benchmark leave-out test set, with accuracy optimization yielding the top-performing model (Appendix Fig. S5).

During training, five proximity labeling features from Gupta et al ciliated condition ('AvgSpec', 'AvgP', 'MaxP', 'Fold_Change',

'BFDR') were deemed to have no predictive value by AutoGluon and were removed. In total, AutoGluon evaluated 13 different base models, and a final ensemble model. The final ensemble model consisting solely of XGBoost with a weight of 1, was the best performer (Dataset EV9). We used this final model on all protein pairs within the feature matrix to predict whether they are interactions or not, including a confidence score. Links to Jupyter notebooks containing modeling training, testing, and prediction generation are provided in the Reagents and Tools Table.

We evaluated the final model using precision-recall analysis on the benchmark leave-out test set and compared its performance to other datasets, including our previous hu.MAP2.0 model (Drew et al, 2021) and BioPlex 3.0 (Huttlin et al, 2021) (Fig. 2A). All pairwise predictions for protein pairs in hu.MAP3.0 are available in Dataset EV7.

## Feature importance analysis

To evaluate which features were most useful in predicting interactions we analyzed our features using the built-in module from AutoGluon. The feature importance module uses permutation importance, which shuffles each feature individually and evaluates the change in model performance (e.g., model accuracy) when making predictions with the perturbed feature matrix. For the feature importance score, positive values describe the drop in performance upon feature perturbation, while negative values indicate a performance gain. We evaluated the importance of our features using the leave-out test set interactions. All importance scores for features can be found in Dataset EV1. We additionally evaluated the predictive power of features when divided into experiment classes. Specifically, we examined features from AP-MS experiments, CF-MS experiments, and features which are from the WMM. To ablate these feature classes, we set all data in these features to zero in the test set feature matrix, calculated confidence scores using the hu.MAP3.0 model and used precision-recall (PR) analysis to evaluate changes in performance (Fig. EV1E).

## Clustering and parameter set selection

To identify clusters within the protein interaction network we used a two-staged clustering approach outlined in Fig. 2B. We first threshold the network based on the confidence score output from the model. To the thresholded network we applied the ClusterOne (Nepusz et al, 2012) algorithm to identify dense regions within the network. For each identified dense region, we applied the MCL (Enright et al, 2002) algorithm to further identify clusters. To determine optimal parameters for cluster identification, we ran parameter sweeps generating clusters with various parameter combinations. Specifically, we used a range of parameters for model confidence score (0.95, 0.9, 0.85, 0.8, 0.7, 0.6, 0.5, 0.4, 0.3, 0.2, 0.1, 0.05, 0.04, 0.03), ClusterOne max overlap (0.6, 0.7), ClusterOne density (0.1, 0.2, 0.3, 0.4), and MCL inflation (1, 2, 3, 4, 5, 7, 9, 11, 15). In addition, after clustering we applied a filter to remove proteins from the resulting clusters which do not have edge weights greater than the model confidence score threshold, which infrequently arose as a result of the MCL algorithm. Finally, we only consider clusters where the number of subunits is ≤100 proteins as larger clusters appeared, after manual inspection, to have disparate biological functions.

For clustering evaluation (Fig. 2C,D), we used the $k$-cliques method, focusing specifically on weighted precision (P_weighted) and weighted recall (R_weighted) as detailed in our previous work (Drew et al, 2021, 2017a). In essence, the $k$-cliques method globally compares clusters to gold standard complexes by evaluating cliques, fully connected subgroups, extracted from the clusters comparing against cliques from the gold standard complexes. This comparison spans all clique sizes from pairs (size 2) to the largest complex or cluster (size $n$). For each clique size, precision and recall are computed. These metrics are then averaged across all clique sizes, with weights assigned based on the number of clusters with size ≥ to the clique size, to reduce the influence of larger clusters on the overall precision and recall.

All cluster sets were analyzed using the $k$-cliques method, with comparisons to the training set of gold standard protein complexes. From this, we manually selected six cluster sets that provided optimal balance between precision and recall, as illustrated in Fig. 2C. These five clusterings were merged into a union set, and the parameters for these clusters are shown in Table 3.

Final evaluation of the selected clusters, we used the $k$-cliques method (Drew et al, 2021) against the gold standard leave-out test set complexes for all individual cluster sets and the union set (Fig. 2D). We additionally compare against previously published Bioplex 3.0 protein complexes. The complete list of hu.MAP3.0 complexes with UniProt identifiers and confidence scores are available in Dataset EV2.

In addition to the complete list of complexes, we provide a reduced set of complexes that minimizes redundancy and preserves the majority of interactions. To generate the reduced set, we calculated pairwise Jaccard similarity coefficients across all complexes. For any pair of complexes with a Jaccard coefficient of 0.7 or greater, indicating at least 70% overlap in subunit composition, we retained only the larger complex for the final reduced set. The resulting reduced set achieves comparable $k$-cliques weighted precision (0.50) and recall (0.51) to the full union set of complexes (Fig. 2D).

## Annotation enrichment analysis for hu.MAP3.0 complexes

For all complexes identified in hu.MAP3.0, we calculated annotation enrichment for GO, Reactome, CORUM, KEGG, and Human Phenotype Ontology (HP) using gProfiler (Reimand et al, 2016) (Dataset EV3). We evaluate complex enrichment by comparing the results against a set of randomly shuffled complexes. In shuffled complexes, protein IDs were randomly reassigned to cluster IDs, maintaining the same size distribution of clusters and preserving the original protein-per-complex distribution. We applied a $p$-value threshold of 0.01 for annotations in all categories and excluded enrichments that are mapped to a single annotated protein. Additionally, we excluded annotations derived via electronic transfer and used the entire set of proteins from the ~25,000 mass spectrometry experiments as the background.

Within this analysis, we also sought to identify uncharacterized proteins within the predicted hu.MAP3.0 complexes to facilitate annotation transfer. We used UniProt annotation scores (www.uniprot.org/help/annotation_score), which range from 1 (least characterized) to 5 (most characterized), to classify proteins within predicted complexes and to identify uncharacterized

**Table 3. Clustering parameters.**

| Clustering | Confidence | Weighted clique precision | Weighted clique recall | F1 score weighted clique | Total clusters | Total proteins | Confidence score threshold | ClusterOne density | ClusterOne overlap | MCL inflation |
|---|---|---|---|---|---|---|---|---|---|---|
| 1 | Extremely high | 0.72 | 0.31 | 0.43 | 5468 | 11221 | 0.95 | 0.2 | 0.7 | 15 |
| 2 | Very high | 0.66 | 0.34 | 0.45 | 5678 | 12049 | 0.95 | 0.3 | 0.7 | 5 |
| 3 | High | 0.63 | 0.39 | 0.48 | 5302 | 11970 | 0.95 | 0.1 | 0.7 | 4 |
| 4 | Moderately high | 0.58 | 0.44 | 0.5 | 5237 | 12346 | 0.95 | 0.1 | 0.7 | 3 |
| 5 | Medium high | 0.53 | 0.48 | 0.5 | 4827 | 12449 | 0.95 | 0.2 | 0.7 | 2 |
| 6 | Medium | 0.47 | 0.53 | 0.5 | 5503 | 13532 | 0.8 | 0.1 | 0.7 | 2 |

proteins. To transfer annotations to previously uncharacterized proteins, complexes were required to meet the following criteria: (1) 50% of complex members must share the same annotation, (2) the annotation must have a corrected $p$-value ≤0.01, (3) the uncharacterized protein must have a UniProt annotation score <4. Annotations were considered from KEGG, GO, CORUM, and Reactome. "CORUM root", "REACTOME root term", and "KEGG root term" were filtered out prior to analysis.

## Covariation analysis with ProteomeHD.2

ProteomeHD.2 is the second version of ProteomeHD, our previously reported protein covariation map (Kustatscher et al, 2019). ProteomeHD.2 will be described in detail in a separate manuscript (Kourtis et al, in preparation). In brief, the workflow for producing ProteomeHD.2 was similar to the one for ProteomeHD, involving the re-processing of 23,000 previously published mass spectrometry raw files deposited in the PRIDE repository (Perez-Riverol et al, 2021). Together, these data covered protein abundance changes in response to 2498 biological perturbations (e.g., drug vs control treatments), all quantified using stable-isotope labeling by amino acids in cell culture (SILAC). To identify proteins with correlated abundance changes across these 2498 conditions, an unsupervised covariation network was constructed using the treeClust algorithm (Kustatscher et al, 2019). In contrast to ProteomeHD, for ProteomeHD.2 an additional supervised machine-learning step was included: the covariation scores, together with a range of edge quality features (e.g., the number of peptides detected per protein or the number of shared samples in which both proteins were detected) were used to train a Random Forest model to distinguish between genuine protein–protein covariation and likely false positive interactions. A combination of protein pairs, known to be functionally related according to STRING, REACTOME or BIOGRID, were used as training data. The resulting Random Forest probabilities were used as final protein covariation score, and were integrated with the hu.MAP3.0 protein–protein interaction scores at the protein-pair level for downstream analysis.

To determine the degree of correlation between all protein pairs' ProteomeHD.2 scores and hu.MAP3.0 scores, we thresholded both scores at 0.2 to correct for the vast majority of non-interacting and non-covarying protein pairs with low scores. Given this threshold, we calculated the Pearson correlation coefficient using SciPy (Virtanen et al, 2020) 'pearsonr' function. To determine the correlation between average ProteomeHD.2 scores and average hu.MAP3.0 scores per complex, we filtered complexes that did not have >80% entries for both datasets. We then used the SciPy (Virtanen et al, 2020) 'pearsonr' function to calculate the Pearson correlation coefficient.

## Structural modeling of uncharacterized proteins

AlphaFold3 web server (Abramson et al, 2024) was used to model the three-dimensional structure of protein interactions in Figs. 4 and 5. Canonical sequences from UniProt were used for each protein. All defaults were used. TMEM167A and IER3IP1 are both identified as transmembrane proteins, therefore 12 copies of palmitic acid were included in the modeling. We use ipTM score to evaluate the model correctness and report those scores in the figures and text for AlphaFold3 models.

## Identification and characterization of mutually exclusive protein subunits

To identify mutually exclusive interactions within hu.MAP3.0 complexes, we examined 136,545 AlphaFold2 structures of pairs of co-complex proteins (Burke et al, 2023; Jänes et al, 2024) (see Reagents and Tools Table). From these dimers, we generated all possible combinations of paired dimers that share a common subunit, resulting in 3,147,494 non-redundant potential 3-mers. We then retained only those 3-mers in which all proteins were present in at least one hu.MAP3.0 complex, yielding 86,514 total 3-mers.

To ensure robust comparisons, we filtered 3-mers based on AlphaFold2 model quality using the pDockQ score (>0.23) and average interface plDDT (IF pLDDT) (>70). A pDockQ score greater than 0.23 has been previously shown to correspond to 70% of structures being well-modeled for a single conformation (Burke et al, 2023). IF pLDDT reflects the confidence of the positioned interface residues, with values >70 considered to be well-modeled (Bryant et al, 2021; Jumper et al, 2021). Additionally, for the AlphaFold2 models from Burke et al, we filtered out pairs that had more than five residue clashes between the two modeled chains for a single pair. This information was only available for Burke et al and we did not apply this filter for the Jänes et al modeled dimers.

To classify two proteins as (1) mutually exclusive or (2) structurally consistent, we used identified 3-mers composed of dimeric AlphaFold2 models that shared a common third subunit. We aligned these structures using the PyMol (Schrödinger, LLC, 2015) align function on the shared subunit and retained only 3-mers with an alignment RMSD < 10 Å. This threshold

accommodates variation in AlphaFold2 models while ensuring good alignment on the shared subunit of both dimers.

To assess whether the two unique proteins are mutually exclusive or structurally consistent, we compared their interfaces with the shared common protein and determined if an interface clash occurred (see Fig. 6 'Workflow'). An interface is defined by residues on the common subunit within 4 Å of the unique proteins. The two models are then aligned on the coordinates of the common protein. An interface clash is determined if the aligned common subunit interface overlaps. Specifically, if the interfaces of the common subunits overlap by >10 interface residues within 4.0 Å, it is considered an interface clash and the pair of unique proteins is considered mutually exclusive. Pairs with 0 clashing interface residues were considered structurally consistent. Due to expected error in AlphaFold2 models, we considered pairs with <10 and >0 clashing interface residues to be inconclusive.

Additionally, we tracked residue clashes between the unique protein chains, regardless of their presence at defined interfaces, to further refine structurally consistent pairs. Clashes between residues of the unique chains could result from poorly modeled regions and may not indicate structural inconsistency. To differentiate genuine clashes from modeling errors, we calculated the average pLDDT of the clashing residues per chain and used this to identify the lower-confidence chain. We then computed the sum of per-residue pLDDT for the clashing residues of the lower-confidence chain. To define a sensible cutoff we compared the distributions between mutually exclusive and structurally consistent pairs. Our reasoning is that mutually exclusive pairs are expected to have chain clashes and provide a useful limit for the minimum acceptable chain clashes. We therefore retained structurally consistent interactions which had a summed chain clash pLDDT <500. This resulted in <2% of mutually exclusive pairs having summed pLDDT values below this threshold.

We further filtered 3-mers for ambiguous cases, including where the shared subunit is known to homomultimerize as identified by AlphaFold2 modeling (Schweke et al, 2024) and therefore may present identical interaction interfaces. We also remove interactions that were inconsistent across multiple common subunits (e.g., mutually exclusive with one common protein but structurally consistent with another). A link for the code repository used to determine mutually exclusive pairs is available in the Reagents and Tools Table.

We calculated annotated enrichment for GO terms (e.g., GO:MF, GO:BP, and GO:CC) of mutually exclusive protein pairs using g:Profiler (Reimand et al, 2016). We took all proteins (i.e., the common and unique subunits) of identified mutually exclusive interactions (3395 proteins) and compared them against all 4131 proteins identified between the mutually exclusive and structurally consistent groups as the background.

We analyzed characteristics of mutual exclusivity, focusing on: (1) Paralogs between mutually exclusive and structurally consistent protein pairs, (2) Covariation of these pairs, (3) Differential tissue expression, and (4) Subcellular compartmentalization. For paralog identification, we used eggNOG (Huerta-Cepas et al, 2019) ortholog groups at the Eukaryota level and determined if both unique proteins in each pair belonged to the same ortholog group. Covariation was assessed using ProteomeHD.2 scores (Kourtis, 2025), which reflect the probability that two proteins covary across cell types and conditions. To evaluate the expression similarity

across tissues, we used RNA-seq data from Human Protein Atlas (Uhlén et al, 2015) (https://www.proteinatlas.org/download/rna_tissue_consensus.tsv.zip) and calculated Pearson's correlation coefficient of normalized RNA expression (nTPM) across tissues. We also analyzed tissue specificity scores (https://www.proteinatlas.org/about/assays+annotation#classification_rna) from the Human Protein Atlas (Uhlén et al, 2015) and compared the number of pairs across categories (Fig. EV5B). Lastly, we compared subcellular localization annotations using Jaccard similarity coefficients between the unique proteins, based on data from the Human Protein Atlas (https://www.proteinatlas.org/download/proteinatlas.tsv.zip) (Fig. EV5C). All of these metrics are annotated within the table of identified mutually exclusive pairs (Dataset EV6).

## Cancer lineage-related protein expression analysis

Cancer lineage-specific protein expression data were extracted from Supplementary Table 2 (Protein matrix) of Gonçalves et al (Gonçalves et al, 2022). Log-scale protein intensities were z-scored for downstream analysis, thereby reflecting protein expression changes relative to their mean abundance across all cell lines. We refer to these z-scores as relative protein abundances. These data were used to create the ALL-IN (Average celL Lineage co-expressIoN) score, designed to reflect the likelihood of a protein pair to be mutually exclusive subunits of a protein complex. To create the ALL-IN score we first calculated four separate sub-scores for all protein pairs annotated as part of the same hu.MAP3.0 complex:

(1) Robust correlation of protein abundance changes across the 949 cell lines, determined using robust correlation (bicor) (Langfelder and Horvath, 2012). Unlike the machine-learning-based similarity metrics used for ProteomeHD, correlation metrics can detect anti-correlation (negative correlation), which is expected for mutually exclusive subunits.

(2) Robust correlation of protein abundance changes after averaging protein intensities by cancer lineage. By averaging values across lineages, a potential bias in subscore 1 stemming from different lineages having different numbers of cell lines, is removed.

(3) A sub-score reflecting very large expression differences. Subscores 1 and 2 fail to detect very large abundance differences, because it is not possible to obtain correlations for samples where a protein was below the detection threshold and thus is not quantified. To overcome this, we first subset the dataset to include proteins that were detected in at least 50 cell lines. The vast majority of proteins in the resulting dataset had z-scores between −2 and 2. We then impute missing values by random sampling from a normal distribution centered around −4 with a standard deviation of 0.3, to reflect that lack of detection was likely caused by low protein abundance. Next, for each protein, we consider the 50 cell lines with the highest and lowest abundance levels (including imputed values), respectively. For each protein pair, we then calculate the difference between the average abundance level (z-score) of protein A in the top 50 and the bottom 50 cell lines of protein B, and vice versa. We combine the values for protein A and B by averaging. The resulting sub-score reflects the degree of differential protein expression between A and B. Unlike sub-scores 1 and 2 this

takes into account missing (imputed) values, i.e., cell lines where one of the two proteins was not detectable.

(4) For each subunit, we calculate the average robust correlation to all other subunits of a complex, separately for each lineage. This results in a correlation coefficient indicating how well each protein correlates with the rest of the complex in each lineage. For each protein pair we then correlate these lineage-specific correlations. The rationale of this sub-score is that, for mutually exclusive subunits expressed in a lineage-specific manner, the dominant subunit in a lineage should correlate well with the rest of the complex in that lineage. By identifying pairs which differ in their lineage-specific correlation with the main complex we can pinpoint possible mutually exclusive pairs.

To combine these four sub-scores into the ALL-IN score for mutually exclusivity, each sub-score was z-scored and the sub-scores averaged. The average score was reversed to be more interpretable as high (i.e., positive) scores will indicate high likelihood of mutual exclusivity. Only protein pairs with at least two out of the four possible sub-scores were included. ALL-IN scores were calculated for all possible protein pairs within hu.MAP3.0 complexes and are provided in Dataset EV10, with the mutually exclusive and structurally consistent protein pairs identified from structurally modeling specifically annotated.

To differentiate between the likelihood of true mutual exclusivity and poor interaction annotation (e.g., spurious interactions) we calculate the Jaccard Interactome score, which evaluates how much the underlying interactome for a given pair overlaps within the hu.MAP3.0 pairwise interactome map. To limit redundancy and ensure confidence of the pairwise interaction, we evaluate only if a protein has a minimum of three interaction partners with a hu.MAP3.0 confidence score of ≥0.8 across the whole dataset. We then calculate the Jaccard similarity coefficient between the interactomes of the two proteins for each pair to obtain our Jaccard Interactome score.

### Overexpression testing of hu.MAP3.0 complexes in cancer lineages

Proteins belonging to confidence 1 hu.MAP3.0 complexes with more than 3 subunits in the cancer lineage dataset, and with less than 80% missing values, were z-scored as indicated in the previous section. The missing values were then imputed with a normal distribution of mean −2 and standard deviation of 0.3, followed by a second round of z-scoring. The subunit relative abundance scores were averaged to the complex-level per cell line and the enrich function from the EnrichIntersect R package (Zhao et al, 2022) was used to calculate the overrepresentation score and *p*-values, which were subsequently FDR-adjusted. Dataset EV11 includes data of protein-level abundance, average complex abundance per cell line, complex expression by lineage, and enrichment scores by lineage.

### Data availability

Both the manually curated set of reference protein complexes and hu.MAP3.0 predicted assemblies are available from the Complex Portal. Data can be accessed either via the website (www.ebi.ac.uk/complexportal), our ftp site (ftp.ebi.ac.uk/pub/databases/intact/

complex/current/) or our REST API (https://www.ebi.ac.uk/intact/complex-ws/). The FTP site contains files in different formats, grouped by species, and for each species there are separate files containing manually curated complexes and machine learning predicted complexes. The Complex Portal is an open source (Apache 2.0), open data (CC0) project, details on https://www.ebi.ac.uk/complexportal/about#license_privacy. Complexes can also be searched and downloaded using the hu.MAP3.0 website (https://humap3.proteincomplexes.org). The complete hu.MAP3.0 complex map, protein interactions with machine learning confidence scores, test and training data, and feature matrix can be found on https://humap3.proteincomplexes.org/download. Software used to generate and analyze the hu.MAP3.0 complex map can be found on GitHub (https://github.com/KDrewLab/protein_complex_maps). Software used to train, test, and make predictions with huMAP3.0 can be found on GitHub (https://github.com/KDrewLab/huMAP3.0_analysis). Machine learning model and feature matrix is also hosted on HuggingFace (https://huggingface.co/datasets/DrewLab/hu.MAP_3.0). Software used to identify mutually exclusive protein pairs and analyze their features can be found on GitHub (https://github.com/KDrewLab/huMAP3.0_analysis). Software related to ProHD2 and ProCAN covariation, and their integration with hu.MAP3.0 can be found on GitHub (https://github.com/kustatscher-lab/HuMAP3_ProHD2). All pairwise ProteomeHD.2 scores are available on Mendeley (https://data.mendeley.com/datasets/76hjnxczz7/2).

The source data of this paper are collected in the following database record: biostudies:S-SCDT-10_1038-S44320-025-00121-5.

## Peer review information

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

## Acknowledgements

This work was supported by grants from the Biotechnology and Biological Sciences Research Council (BBSRC) - US National Science Foundation Directorate for Biological Sciences (NSF/BIO) [BB/X002179/1][BB/X002683/1][NSF2314278], National Human Genome Research Institute (NHGRI), Office of Director [OD/DPCPSI/ODSS]; National Institute of Allergy and Infectious Diseases (NIAID), National Institute on Aging (NIA), National Institute of General Medical Sciences (NIGMS), National Institute of Diabetes and Digestive and Kidney Diseases (NIDDK), National Eye Institute (NEI), National Cancer Institute (NCI), National Heart, Lung, and Blood Institute (NHLBI) of the National Institutes of Health [U24HG007822] and European Molecular Biology Laboratory (EMBL) core funds. GK is supported by an MRC Career Development Award (MR/T03050X/1) and a Royal Society Research Grant (RGS\R2\212303). KD and SNF are supported by National Institute of Child Health and Human Development (R00 HD092613). We thank Eliot Ragueneau for artwork design.

## Author contributions

**Samantha N Fischer**: Conceptualization; Resources; Data curation; Software; Formal analysis; Validation; Investigation; Visualization; Methodology; Writing—original draft; Writing—review and editing. **Erin R Claussen**: Conceptualization; Data curation; Software; Formal analysis; Validation; Investigation; Visualization; Methodology; Writing—original draft; Writing—review and editing. **Savvas Kourtis**: Conceptualization; Resources; Data curation; Software; Formal analysis; Validation; Investigation; Visualization; Methodology; Writing—original draft; Writing—review and editing. **Sara Sdelci**: Software; Funding acquisition; Writing—review and editing. **Sandra Orchard**: Conceptualization; Resources; Data curation; Software; Supervision; Funding acquisition; Validation; Investigation; Visualization; Writing—original draft; Project administration; Writing—review and editing. **Henning Hermjakob**: Conceptualization; Resources; Data curation; Software; Supervision; Funding acquisition; Validation; Investigation; Visualization; Writing—original draft; Project administration; Writing—review and editing. **Georg Kustatscher**: Conceptualization; Resources; Data curation; Software; Formal analysis; Supervision; Funding acquisition; Validation; Investigation; Visualization; Methodology; Writing—original draft; Project administration; Writing—review and editing. **Kevin Drew**: Conceptualization; Resources; Data curation; Software; Formal analysis; Supervision; Funding acquisition; Validation; Investigation; Visualization; Methodology; Writing—original draft; Project administration; Writing—review and editing.

Source data underlying figure panels in this paper may have individual authorship assigned. Where available, figure panel/source data authorship is listed in the following database record: biostudies:S-SCDT-10_1038-S44320-025-00121-5.

## Disclosure and competing interests statement

The authors declare no competing interests.

# Expanded View Figures

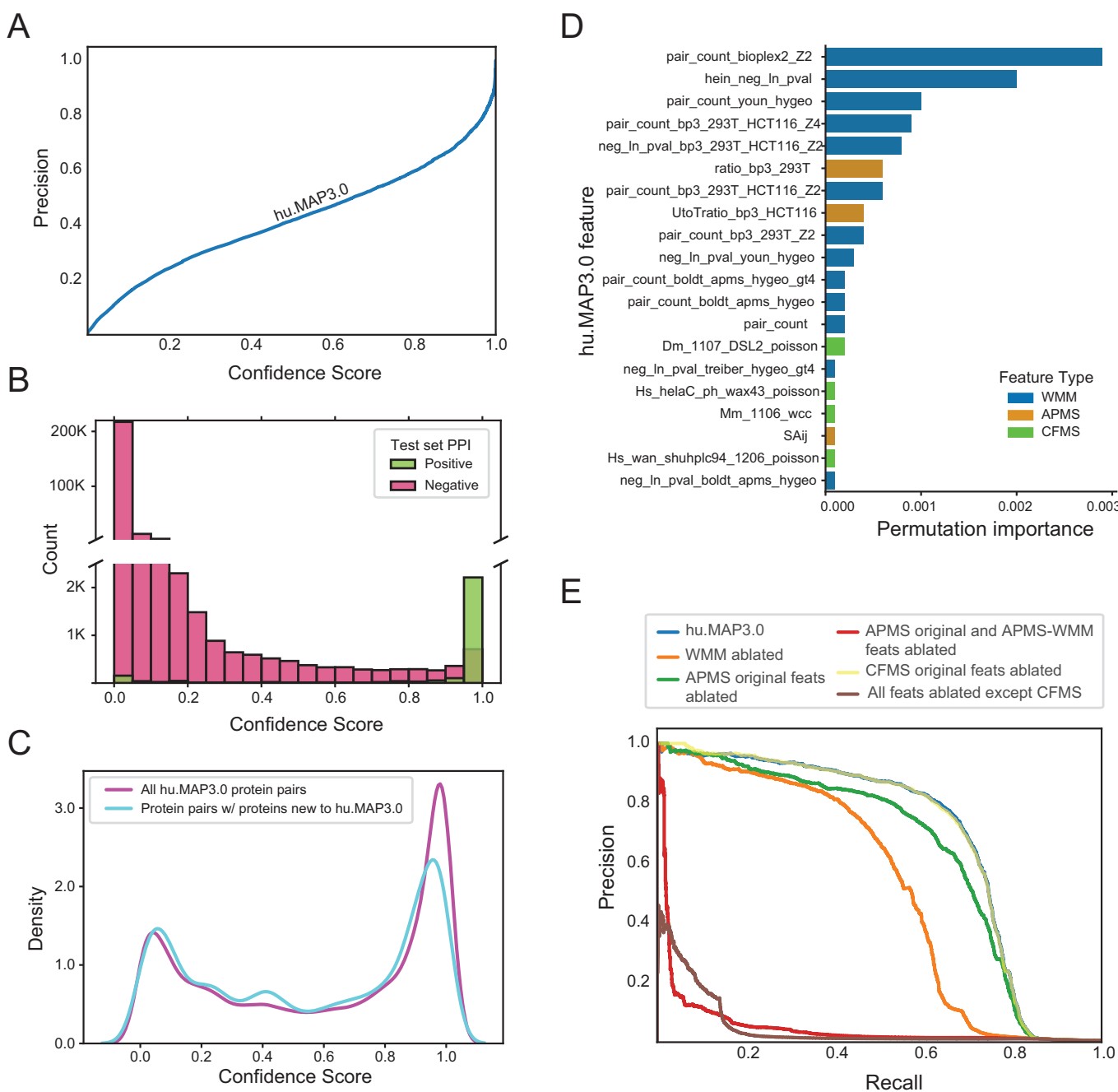

**Figure EV1.   Characterization of hu.MAP3.0 model performance.**

(**A**) Line plot depicting the relationship between precision and the hu.MAP3.0 model confidence score on the test set interactions, illustrating that increases in confidence score correspond with increased precision. (**B**) Distribution of positive and negative test set interactions across model confidence scores. The histogram highlights the separation of these interactions. (**C**) Distribution of hu.MAP3.0 confidence scores for all pairs (violet) and pairs with protein new to hu.MAP3.0 (light blue). (**D**) Autogluon feature importance for top 20 evidence features used in hu.MAP3.0 machine learning classifier. Colors represent feature categories including Weighted Matrix Model (WMM), affinity purification based features (APMS), and co-fractionation (CFMS). (**E**) Precision recall curves for different feature category ablation tests.

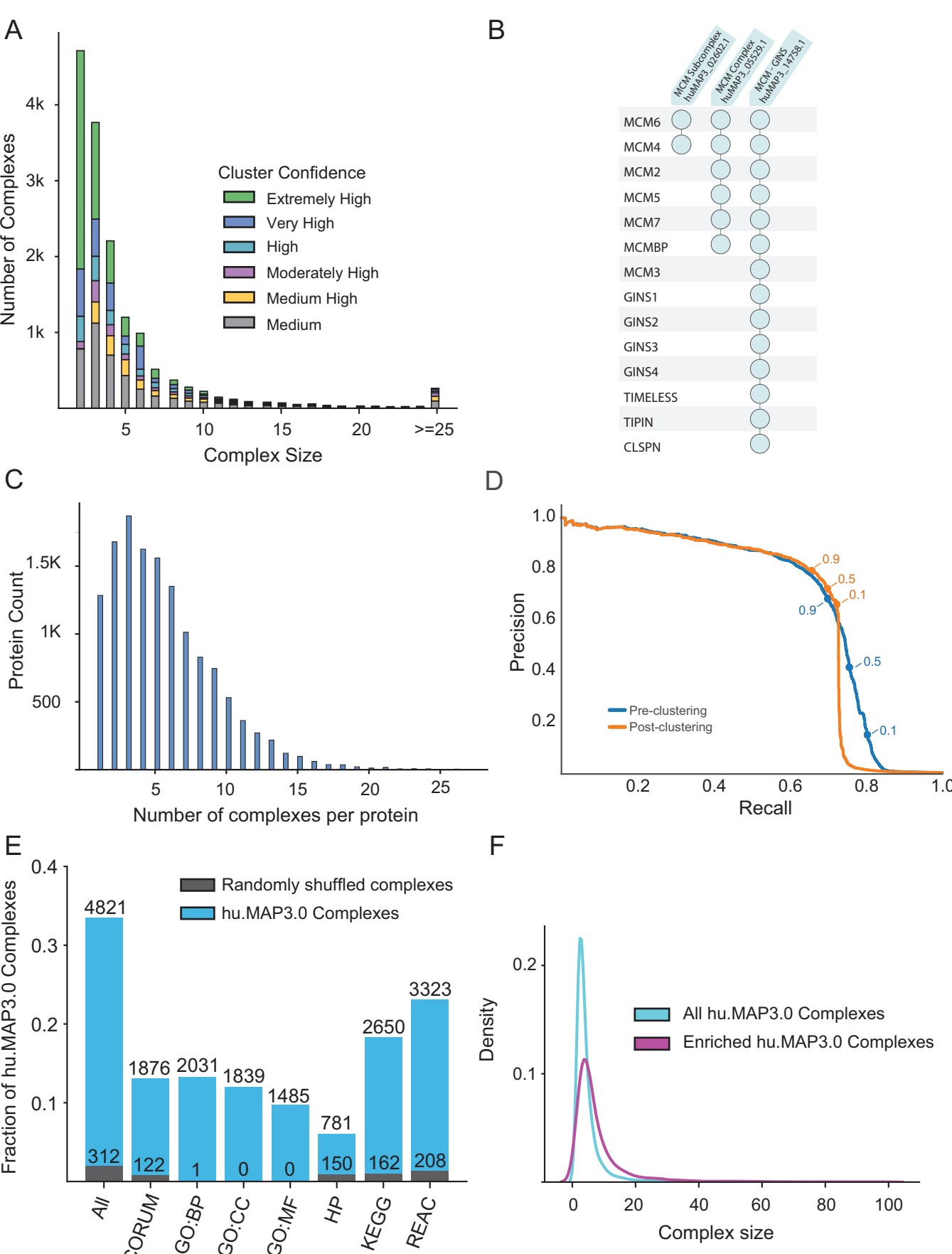

◀ **Figure EV2. Clustering size distribution and annotation enrichment.**

(A) Distribution of hu.MAP3.0 complex size. Colors in stacked bars represent cluster confidence. (B) Subunit composition of MCM complex hierarchy. (C) Distribution of the number of complexes assigned to individual proteins. (D) Precision-Recall plot for the hu.MAP3.0 model on test set interactions pre- and post-clustering, with confidence score thresholds marked as circles, demonstrating clear separation along the curve. (E) Annotation enrichment of hu.MAP3.0 complexes using g:Profiler. Bars represent the number of complexes enriched for GO, KEGG, CORUM, Reactome, or Human Phenotype Ontology (HP) annotations that pass a corrected *p*-value of 0.01. Random represents enriched protein sets when complex membership is shuffled for all complexes. (F) Distribution of hu.MAP3.0 complex size for all complexes (light blue) and complexes enriched with annotation (violet).

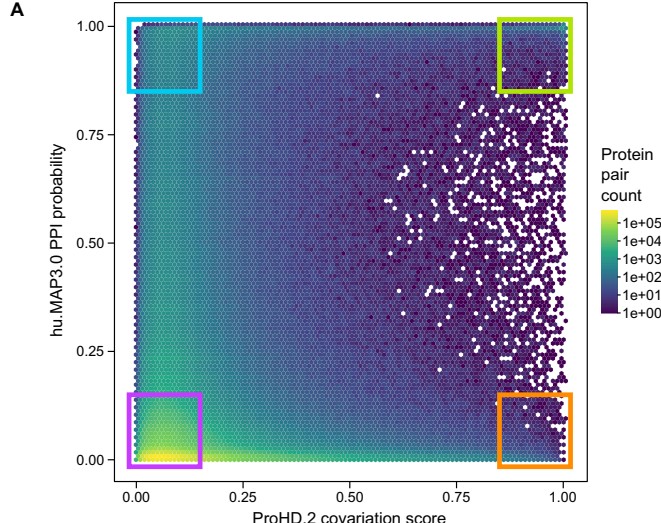

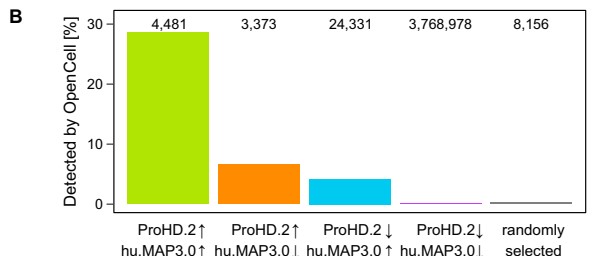

**Figure EV3. Comparison of ProteomeHD.2, hu.MAP3.0, and OpenCell.**

(A) Scatterplot of protein pairs' ProteomeHD.2 covariation score (x-axis) and hu.MAP3.0 confidence score (y-axis). Four quadrants are highlighted where yellow = high in hu.MAP3.0, high in ProteomeHD.2, blue = high in hu.MAP3.0, low in ProteomeHD.2, orange = low in hu.MAP3.0, high in ProteomeHD.2, and purple = low in both. (B) Comparison of four quadrants in (A) to an orthogonal protein interaction dataset, OpenCell, not included in either hu.MAP3.0 nor ProteomeHD.2. Bars represent the percent detected in OpenCell. Number of protein pairs per category are shown above the bars.

A  COP9 Signalosome
(humap3_05041.1)

hu.MAP3.0 Score

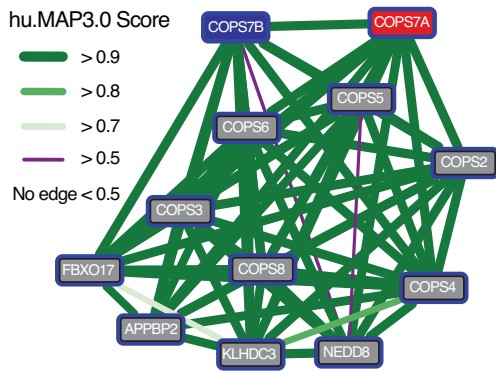

B

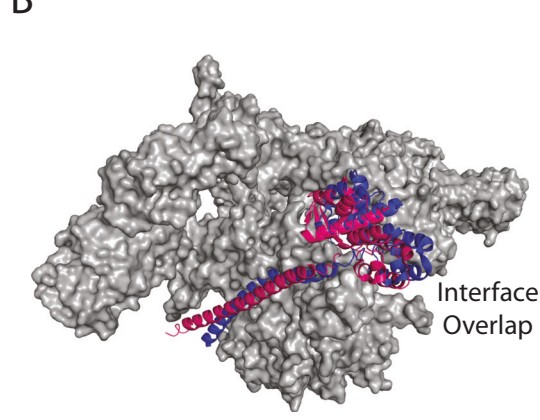

Interface
Overlap

C

ProteomeHD.2

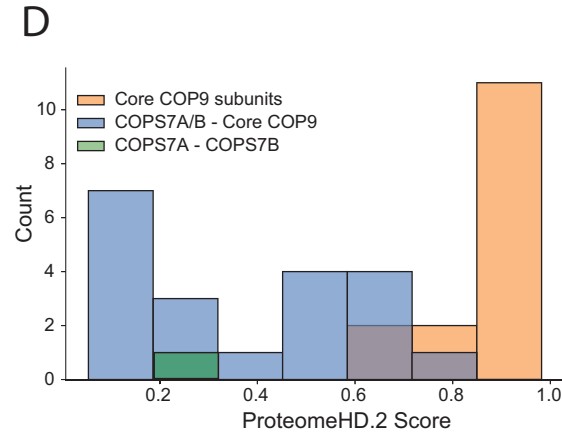

D

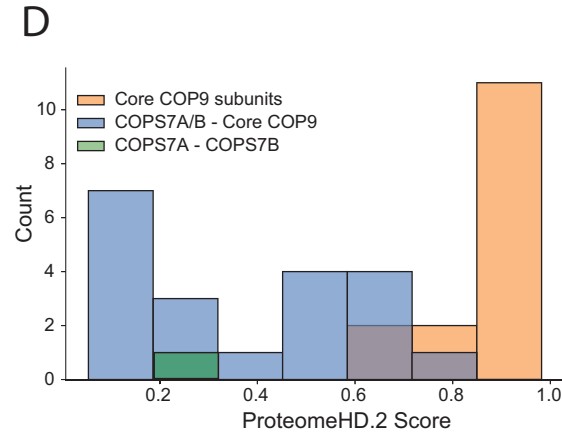

E  BRISC/BRCA1-A Complex
(humap3_08508.1)

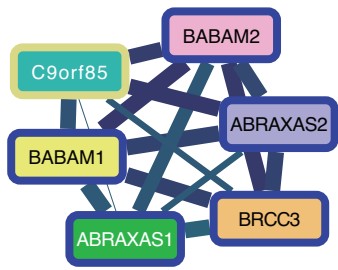

F

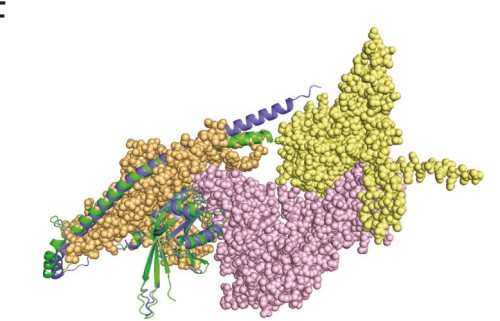

G  BRISC Complex

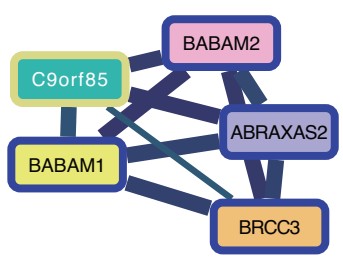

H  BRCA1-A Complex

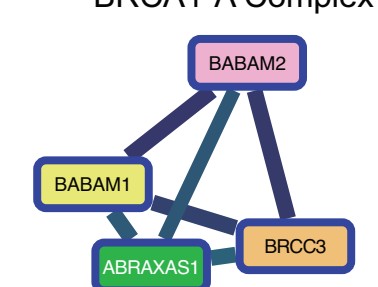

◀    **Figure EV4.   Example complexes with mutual exclusive subunits.**

(A) Co-complex interaction network of the COP9 Signalosome shows high interconnectivity of core subunits and peripheral subunits FBXO17, APPBP2, KLHDC3, and NEDD8. (B) Structure of COP9 Signalosome core subunits show overlapping interfaces of COPS7A and COPS7B with other core subunits (PDB id: 4D10, 6R6H) pointing to these subunits being mutually exclusive. (C) Protein covariation network of COP9 Signalosome from ProteomeHD.2. Core subunits of the network are highly co-expressed while COPS7A and COPS7B are less co-expressed to core subunits. (D) Distributions of covariation between core COP9 subunits (orange), COPS7A or COPS7B and core COP9 subunits (blue), and the covariation between COPS7A and COPS7B (green). (E) The BRISC/BRCA1-A Complex highlights the identification of two mutually exclusive proteins and the annotation of an uncharacterized protein. Blue border of protein subunits represents a known member of the complex; the yellow border represents an uncharacterized protein. Edge weights represent hu.MAP3.0 confidence score. (F) AlphaFold3 model of known subunits of complex. The interfaces of ABRAXAS1 and ABRAXAS2 with BRCC3 overlap and therefore are determined to be mutually exclusive subunits. (G, H) BRISC/BRCA1-A Complex split into two complexes which appropriately separates mutually exclusive subunits and places the uncharacterized protein, C9ORF85, with the BRISC complex.

                                                                  

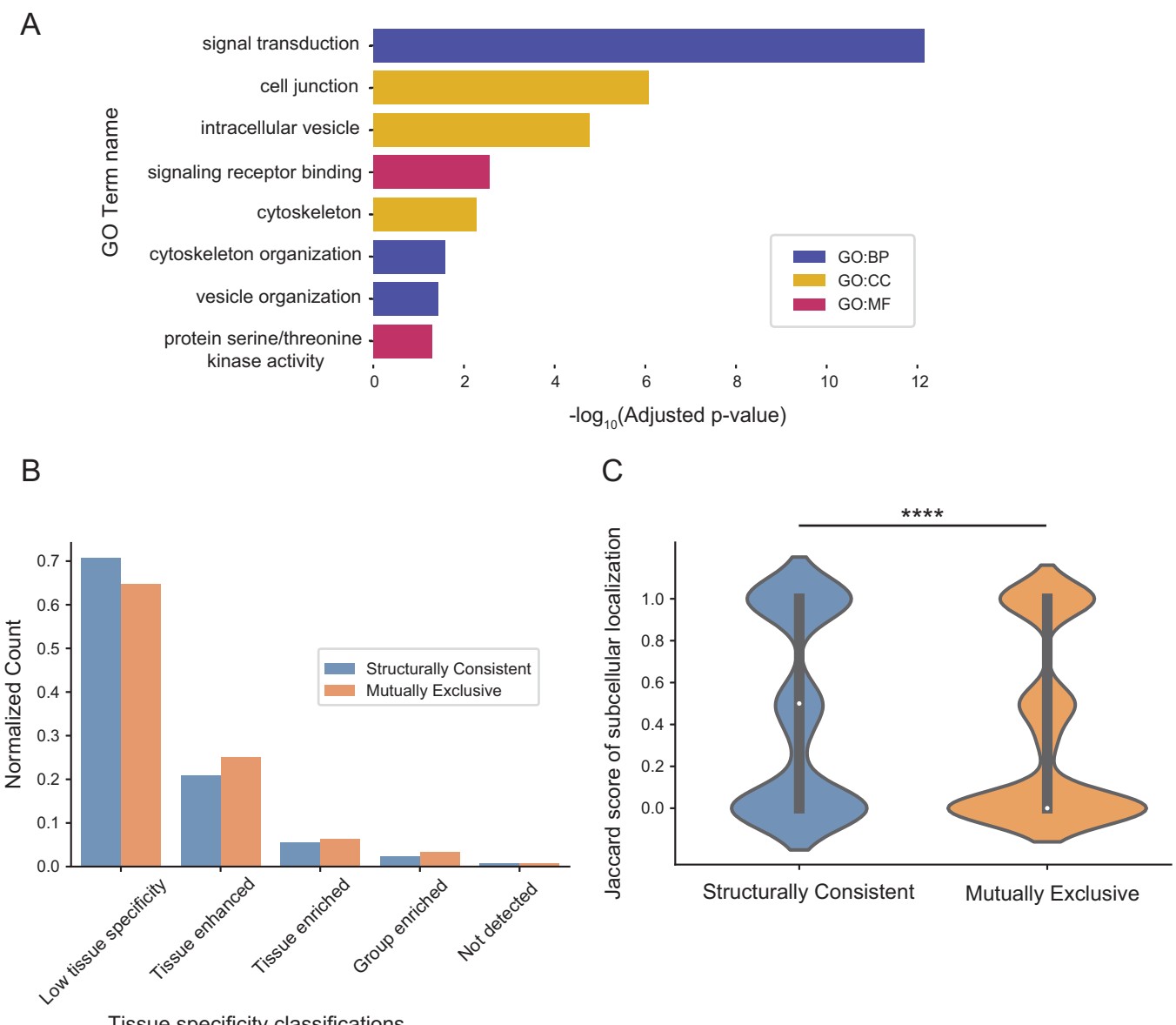

**Figure EV5. Characterization of mutually exclusive interactions.**

(A) Barplot displaying gene ontology annotation enrichments for molecular function (MF), biological process (BP), and cellular compartment (CC) for identified mutually exclusive proteins. (B) Barplot showing the normalized count of proteins from mutually exclusive or structurally consistent protein pairs in different Human Protein Atlas tissue expression classifications. (C) Violin plot showing the distribution of Jaccard similarity of subcellular localizations between proteins in mutually exclusive ($N = 4031$) and structurally consistent ($N = 1830$) protein pairs (two-sided Mann-Whitney $U$-Test, ****$P < 0.0001$, $P = 4.48 \times 10^{-13}$). Violin shape indicates density of data points, with wider sections indicating higher concentration of values. The upper and lower bounds of the inner box indicate the quartiles of the distribution with first (Q1, 25th percentile) and (Q3, 75th percentile), respectively. Here, the first and third quartile are the minimum and maximum values, respectively. The inner dot indicates the median.

