## [Peer Review File · Molecular Systems Biology]

hu.MAP3.0: Atlas of human protein complexes by integration of > 25,000 proteomic experiments

Samantha Fischer, Erin Claussen, Savvas Kourtis, Sara Sdelci, Sandra Orchard, Henning Hermjakob, Georg Kustatscher, and Kevin Drew

Corresponding author(s): Kevin Drew (ksdrew@uic.edu)

Review Timeline:

Submission Date:	22nd Nov 24
Editorial Decision:	7th Jan 25
Revision Received:	19th Mar 25
Editorial Decision:	18th Apr 25
Revision Received:	7th May 25
Accepted:	9th May 25

Editor: Jingyi Hou

Transaction Report:

7th Jan 2025

Manuscript Number: MSB-2024-12769

Title: hu.MAP3.0: Atlas of human protein complexes by integration of > 25,000 proteomic experiments

Author: Samantha Fischer

Erin Claussen

Savvas Kourtis

Sara Sdelci

Sandra Orchard

Henning Hermjakob

Georg Kustatscher

Kevin Drew

Dear Dr Drew,

Thank you for submitting your work to Molecular Systems Biology. First of all, I apologize for the somewhat slow process, which was due to the late arrival of the reviewers' reports. We have now heard back from the three reviewers who agreed to evaluate your manuscript. As you will see from the reports below, the reviewers find the study interesting. They raise, however, a series of concerns, which we would ask you to address in a major revision.

The reviewers' recommendations are relatively clear, so there is no need to reiterate the points listed below. All the issues raised by the reviewers need to be satisfactorily addressed. As you may already know, our editorial policy allows in principle a single round of major revision, and it is therefore essential to provide responses to the reviewers' comments that are as complete as possible. Please feel free to contact me in case you would like to discuss in further detail any of the issues raised by the reviewers.

On a more editorial level, we would ask you to address the following issues:

- Please provide a .docx formatted version of the manuscript text (including legends for main figures, EV figures and tables). Please make sure that the changes are highlighted to be clearly visible.
- Please provide individual production quality figure files as .eps, .tif, .jpg (one file per figure).
- Please provide a .docx formatted letter INCLUDING the reviewers' reports and your detailed point-by-point responses to their comments. As part of the EMBO Press transparent editorial process, the point-by-point response is part of the Review Process File (RPF), which will be published alongside your paper.
- Please note that all corresponding authors are required to supply an ORCID ID for their name upon submission of a revised manuscript.
- We replaced Supplementary Information with Expanded View (EV) Figures and Tables that are collapsible/expandable online (see examples in <http://msb.embopress.org/content/11/6/812>). A maximum of 5 EV Figures can be typeset. EV Figures should be cited as 'Figure EV1, Figure EV2' etc... in the text and their respective legends should be included in the main text after the legends of regular figures.

Additional Tables/Datasets should be labeled and referred to as Table EV1, Dataset EV1, etc. Legends have to be provided in a separate tab in case of .xls files. Alternatively, the legend can be supplied as a separate text file (README) and zipped together with the Table/Dataset file.

For the figures and tables that you do NOT wish to display as Expanded View figures, they should be bundled together with their legends in a single PDF file called *Appendix*, which should start with a short Table of Content. Each legend should be below the corresponding Figure/Table in the Appendix. Appendix figures and tables should be referred to in the main text as: "Appendix Figure S1, Appendix Figure S2, Appendix Table S1" etc. See detailed instructions regarding expanded view here: <https://www.embopress.org/page/journal/17444292/authorguide#expandedview>.

- Before submitting your revision, primary datasets (and computer code, where appropriate) produced in this study need to be deposited in an appropriate public database (see <http://msb.embopress.org/authorguide-dataavailability> <https://www.embopress.org/page/journal/17444292/authorguide#dataavailability>). Please remember to provide a reviewer password if the datasets are not yet public. The accession numbers and database should be listed in a formal "Data Availability" section (placed after Materials & Method) that follows the model below (see also <https://www.embopress.org/page/journal/17444292/authorguide#dataavailability>). Please note that the Data Availability Section is restricted to new primary data that are part of this study.

Data availability

- At EMBO Press we ask authors to provide source data for the main figures. Our source data coordinator will contact you to discuss which figure panels we would need source data for and will also provide you with helpful tips on how to upload and organize the files.

- Our journal encourages inclusion of *data citations in the reference list* to directly cite datasets that were re-used and obtained from public databases. Data citations in the article text are distinct from normal bibliographical citations and should directly link to the database records from which the data can be accessed. In the main text, data citations are formatted as follows: "Data ref: Smith et al, 2001". In the Reference list, data citations must be labeled with "[DATASET]". A data reference must provide the database name, accession number/identifiers and a resolvable link to the landing page from which the data can be accessed at the end of the reference. Further instructions are available at .

- We updated our journal's competing interests policy in January 2022 and request authors to consider both actual and perceived competing interests. Please review the policy <https://www.embopress.org/competing-interests> and update your competing interests if necessary.

Please use the heading "Disclosure statement and competing interests".

- All Materials and Methods need to be described in the main text using our 'Structured Methods' format. According to this format, the Methods section includes a Reagents and Tools Table (listing key reagents, experimental models, software and relevant equipment and including their sources and relevant identifiers) followed by a Methods and Protocols section describing the methods, ideally using a step-by-step protocol format. The aim is to facilitate adoption of the methodologies across labs. Please download and fill our Reagents and Tools Table template (.docx), which you can find in our author guidelines: <https://www.embopress.org/page/journal/17444292/authorguide#structuredmethods>.

- Regarding data quantification:

Please ensure to specify the name of the statistical test used to generate error bars and P values, the number (n) of independent experiments (please specify technical or biological replicates) underlying each data point and the test used to calculate p-values in each figure legend. Discussion of statistical methodology can be reported in the materials and methods section, but figure legends should contain a basic description of n, P and the test applied.

Graphs must include a description of the bars and the error bars (s.d., s.e.m.).

- Please provide a "standfirst text" summarizing the study in one or two sentences (approximately 250 characters, including space), three to four "bullet points" highlighting the main findings and a "synopsis image" (550px width and 400-600 px height, PNG format) to highlight the paper on our homepage.

Here are a couple of examples:

<https://www.embopress.org/doi/10.15252/msb.20199356>

<https://www.embopress.org/doi/10.15252/msb.20209475>

<https://www.embopress.org/doi/10.15252/msb.209495>

When you resubmit your manuscript, please download our CHECKLIST (<https://www.embopress.org/pb-assets/embo-site/EMBO%20Press%20Author%20Checklist-1642513524327.xlsx>) and include the completed form in your submission.

Please note that the Author Checklist will be published alongside the paper as part of the transparent process (<https://www.embopress.org/page/journal/17444292/authorguide#transparentprocess>).

If you feel you can satisfactorily deal with these points and those listed by the referees, you may wish to submit a revised version of your manuscript. Please attach a covering letter giving details of the way in which you have handled each of the points raised by the referees. A revised manuscript will be once again subject to review and you probably understand that we can give you no guarantee at this stage that the eventual outcome will be favorable.

I look forward to receiving your revised manuscript soon.

Sincerely,
Jingyi

Jingyi Hou, PhD
Senior Editor
Molecular Systems Biology

We realize that it is difficult to revise to a specific deadline. In the interest of protecting the conceptual advance provided by the work, we recommend a revision within 3 months (7th Apr 2025). Please discuss the revision progress ahead of this time with the editor if you require more time to complete the revisions. Use the link below to submit your revision:

IMPORTANT: When you send your revision, we will require the following items:

1. the manuscript text in LaTeX, RTF or MS Word format
2. a letter with a detailed description of the changes made in response to the referees. Please specify clearly the exact places in the text (pages and paragraphs) where each change has been made in response to each specific comment given
3. three to four 'bullet points' highlighting the main findings of your study
4. a short 'blurb' text summarizing in two sentences the study (max. 250 characters)
5. a 'thumbnail image' (550px width and max 400px height, Illustrator, PowerPoint or jpeg format), which can be used as 'visual title' for the synopsis section of your paper.
6. Please include an author contributions statement after the Acknowledgements section (see <https://www.embopress.org/page/journal/17444292/authorguide>)
7. Please complete the CHECKLIST available at (<https://bit.ly/EMBOPressAuthorChecklist>). Please note that the Author Checklist will be published alongside the paper as part of the transparent process (<https://www.embopress.org/page/journal/17444292/authorguide#transparentprocess>).
8. When assembling figures, please refer to our figure preparation guideline in order to ensure proper formatting and readability in print as well as on screen:

See also figure legend guidelines: <https://www.embopress.org/page/journal/17444292/authorguide#figureformat>

9. Please note that corresponding authors are required to supply an ORCID ID for their name upon submission of a revised manuscript (EMBO Press signed a joint statement to encourage ORCID adoption). (<https://www.embopress.org/page/journal/17444292/authorguide#editorialprocess>)
Currently, our records indicate that the ORCID for your account is 0000-0002-1260-4413.

Link Not Available

11. Include a Reagents and Tools Table as part of the Methods section, which can be downloaded from our author guidelines (<https://www.embopress.org/page/journal/17444292/authorguide#structuredmethods>)

*** PLEASE NOTE *** As part of the EMBO Press transparent editorial process initiative (see our Editorial at <https://dx.doi.org/10.1038/msb.2010.72>), Molecular Systems Biology publishes online a Review Process File with each accepted manuscripts. This file will be published in conjunction with your paper and will include the anonymous referee reports, your point-by-point response and all pertinent correspondence relating to the manuscript. If you do NOT want this File to be published, please inform the editorial office at msb@embo.org within 14 days upon receipt of the present letter.

Reviewer #1:

The paper "hu.MAP3.0: Atlas of human protein complexes by integration of >25,000 proteomic experiments" presents a

comprehensive map of human protein complexes derived from the integration of over 25,000 mass spectrometry experiments using advanced machine learning workflows. This resource identifies more than 15,000 protein complexes, encompassing approximately 75% of the human proteome. The study highlights the accuracy and scope of hu.MAP3.0 compared to prior maps and provides novel insights, including the identification of co-varying complexes, mutually exclusive subunits, and functional predictions for understudied proteins. The dataset is enriched with structural insights from AlphaFold modeling and is made publicly accessible via EMBL-EBI's Complex Portal and a dedicated web interface. This work offers a valuable tool for understanding protein interactions and their implications in biology and disease.

One of the major achievements is the use of AF2-multimer to model mutually exclusive interactions partners - this is really novel.

Another major achievement is the use of co-variation both within and between complexes.

In summary this paper presents the state of art data for protein complexes. However, it is still not perfect as for instance JUNO-IZUMO1 is missing.

I could reach humap3.proteomcomplexes.org the other day but today it is unreachable. Perhaps it is a good idea to check the stability of the website. This also meant that I could not check how easy it was to download or use the data (also humap2.proteincomplexes.org was only reachable via HTTP not HTTPS the other day - this should be fixed)

I would urge the authors to make all data FAIR compliant - not clear if it is - in particular ProteomeHD.2 scores are only available upon request - this is not OK.

PS. The submission of manuscripts with all the figures at the end really makes it much harder for this reviewer to read it. This is something the journal should not allow. For every time a figure is mentioned I need to scroll down 20 pages to find the legend and then another 20 pages to find the figure, and finally try to find my way back. In the future I will not review papers formatted in this way.

Reviewer #2:

In this manuscript Fischer and colleagues describe the integration of a very large compendium of proteomics datasets, to generate a high confidence human protein-protein interaction network and list of protein complexes. This is an expansion of the previous studies done in 2017 (Drew et al. MSB 2017) and 2021 (Drew et al. 2021 MSB). The previous versions of this work have been very useful to the scientific community as it can be seen by the number of citations these papers have accumulated. The increase in data incorporated is primarily driven by the addition of the BioPlex 3 project dataset. Despite it being the addition of a single study it corresponds to a very large number of new experiments on its own. Therefore, this corresponds to a very useful integration effort that produces what is likely to be the current largest high confidence integrated dataset of human protein interactions and complexes.

The authors study this resource to provide, for example, functional annotation to understudied proteins as well for comparing it with a very large protein covariation analysis (ProteomeHD.2). This dataset is not yet published but is briefly described in the methods as consisting of co-abundance based estimates of protein associations derived from the reanalysis of nearly 2500 perturbation experiments done in human cells. Given the importance of this covariation dataset for the study that is here under review, I think this data would need to be made available in some form by the time of publication. The co-abundance analysis was used to find cases where proteins complex subunits do not covary, which includes examples of complexes that are mutually exclusive. The authors study mutual exclusivity in more detail by incorporating predicted structural models and in addition study the patterns of co-abundance of proteins in cancer cell proteomics datasets.

In summary, I think the work is a very useful data integration effort that creates a useful resource of a large community of scientists interested in human protein interactions. The methods are clear and data and predictions are readily available for re-use. The combined analysis of protein complexes and co-variation is very interesting as well as the study of mutual exclusive protein complexes. I have only a series of minor concerns that I hope the authors could consider:

1 - The ProteomeHD.2 dataset is critical for the work described here and anyone trying to reproduce the results would need to have access to it. I understand that there is a considerable amount of work behind ProteomeHD.2 but the authors would need to find a way to make that dataset available before publication.

2 - The procedure for the machine-learning model for data-integration is well described but it is not clear in the end if there is any regularization involved or if the final model ends up using all features. The authors could provide as well a list of features with some model weight estimates so that it is clear which features end up being most useful in end result.

3 - A very large number of complexes ends up not being annotated with any gene set. Is this mostly to do with the fact that most complexes are very small ? If the authors exclude the complexes that are <3 or <4 subunits, what is the fraction of non-annotated complexes ? Are there large >10 subunit complexes that do not have any predicted functional annotation ? Related to this, it would be useful to mention briefly the size distribution of complexes in the main text (shown in Figure S2).

4 - I was surprised that the protein complex confidence didn't correlate much with the co-abundance data. It would be useful to also compare directly the protein-protein interaction scores. Is there a general correlation between protein-protein interaction confidence and protein covariation scores ? At the complex level, instead of confidence bin as in Fig S4, what is the correlation between average interaction confidence (for a given complex) and the corresponding average covariation score ?

5 - Regarding the structural analysis and mutually exclusivity, which I really enjoyed, there is a much larger dataset of protein structure predictions in one of the cited studies (Jänes et al. biorxiv 2024). Why not use those models as well to greatly expand the coverage ?

6 - Looking through the scored protein pairs in Table S7, I find around 160 thousand pairs, some of which have very low probabilities (0.0001). The manuscript mentions that the data covers 13,769 human proteins so the space of possible pairs among the proteins with data is many orders of magnitude larger than the 160k values. Figure 1C shows the number of interactions in humap3 as those 160k pairs but so I assume the Table S7 to be the correct number of defined interactions. Are the probability values in Table S7 correct ? What is the relationship between the probability value and model accuracy etc ?

7 - Although this is a subjective opinion, I felt that there were parts of the manuscript that could be shortened without impacting on the messages. This includes: some of initial ML model building and clustering; the examples in "Identification of mutually exclusive subunits in hu.MAP3.0".

8 - Some the methods are described as running some scripts (Reduce redundancy of benchmark complexes and Generating test and train sets). This should be instead the description of the algorithm/steps that are performed by those scripts.

Reviewer #3:

Fischer et al. characterize the complexome of the human proteome. The work is an extension of their previous work Hu.Map2. Hu.Map3 increases both the number of complexes characterized (15,000 against 7,000) and the coverage of the proteome (13,800 vs 10,000 proteins). As in their previous work, a machine learning framework is used to score interactions from thousands of MS-based pull-down and co-fractionation experiments as well as from other resources such as proximity information. The integration of the BioPlex3 resource also serves as an important source of information.

The work builds on a solid resource and extends it further. It will be of immense value to the community. It also adds a novel perspective with a comparison between physical interactions and co-expression information, as inferred from ProteomeHD.2. The original analyses of mutually exclusive subunits and cancer-associated subunits also illustrate ways in which this resource can be used.

Several points could nevertheless be improved, in particular with respect to the presentation of the results and the mutually exclusive subunits, as described in the comments below.

Comments:

1. P.2: This final step results in 15,326 complexes covering 13,769 human proteins and consisting of 159,451 total scored protein interactions. These numbers indicate redundancy among complexes, and it would be valuable to provide some ideas/statistics regarding this redundancy, e.g., the distribution of the number of complexes a protein appears in, and what these complexes correspond to biologically.

2. P.4 and Fig S2: According to the plot, the larger the complexes are, the less reliable they become. The authors state "This is likely due to core members of complexes being confidently identified in high confidence levels and additional auxiliary subunits added to complexes in lower confidence clusters." This could be simply tested/analyzed.

3. P.22 "Additionally, we ensured that AlphaFold2 models had <5 overlapping residues between protein chains." I assume that the authors are referring to residue clashes. As the wording "overlapping residues" is also used by the authors to refer to "shared interfaces residues", this paragraph is not clear and should be re-written/better explained.

4. P.22 "Interface residues between each protein were tested for overlap (residue atoms < 4.0 Å). Pairs with >10 overlapping interface residues were considered mutually exclusive. Pairs with 0 overlapping interface residues were considered structurally consistent. Due to expected error in AlphaFold2 models, we considered pairs with < 10 and > 0 overlapping interface residues inconclusive." Although this will probably not change the conclusions, it could make sense to use a definition of overlap that is a function of interface size (e.g., >5 or 10%) rather than an absolute number (10 residues).

5. P.22 "We tracked residue overlap between unique protein chains, regardless of their presence at defined interfaces, to further

filter the structurally consistent pairs. Overlapping could result from clashes in poorly modeled regions and does not necessarily indicate structural inconsistency. Therefore, we excluded structurally consistent pairs which had > 100 overlapping residues between their unique protein chains." - here as well, if the goal is to filter out clashing residues, it would be clearer to write it explicitly. It would also be more accurate to conduct the analyses on "high confidence regions", e.g., those with a pLDDT above a given cut-off for example. I do not see this as a very critical point as conclusions are unlikely to change, but it could improve the results.

6. Since this work builds on and improves on hu.MAP2, it would be valuable to add a section in the methods where the main differences in the pipeline are mentioned. This would summarize both methodological differences (cut-off values, data sources, number of experiments used as input, number of features used for classification/learning) as well as output differences (number of proteins, PPIs, complexes, etc).

7. Along the same lines, an important result of this work is the larger coverage with almost 4,000 additional proteins. Therefore, this specific set should be highlighted/described better in the figures, and it would give the readers a more precise idea of their characteristics. For instance, are we equally confident about those proteins' interactions? If not, where is the confidence range situated when compared to the rest of the dataset? (a distribution of confidence score for old vs newly added proteins could be shown), etc.

8. The limitations of the current approach are not explored or explained in sufficient detail. To help with this, it would be useful to add a panel characterizing (e.g., through GO term enrichment and specific examples) the FP (negatives with score >0.9) and FN (positives with score <0.1) that are seen in the barplot Figure S1D.

9. In relation to the statement "However, we find that the degree of complex covariation correlates only mildly with hu.MAP3.0 confidence levels (Fig S4)" - I would suggest citing the following review, which discusses this concept extensively:
<https://pubmed.ncbi.nlm.nih.gov/24997301/>

10. It appears that mutually exclusive pairs show weaker co-expression than non-mutually exclusive ones, as one could expect (Fig 6K,L). Later on, it is shown that homomers also bring information with respect to such mutually exclusive subunits (Fig. 7B), so it is not clear why homomer information is not included in the analysis presented in Fig. 6. Related to this point, I would suggest using a recent comprehensive resource on human homomers: <https://pubmed.ncbi.nlm.nih.gov/38325366/> - It would also be appropriate to cite what is to my knowledge the first attempt at mapping PPIs onto similar vs distinct interfaces (<https://pubmed.ncbi.nlm.nih.gov/17185604/>)

11. A limit to the structural analysis should be noted: the pLDDT should ideally not be used to assess the quality of a complex because it reflects on subunit quality rather than interaction quality (the latter requires PAE information). I understand that the authors cannot use PAE-derived metrics to filter out models because the dataset they downloaded does not provide this info, but a sentence could be added to note and explain this discrepancy with other AlphaFold analyses in the paper where they used the ipTM score (readers may not understand why the ipTM is used in certain analyses and not others).

Language / clarity related comments:

* Abstract. "Unfortunately, we lack the subunit composition for all human protein complexes" - this sentence can be interpreted as a COMPLETE lack of knowledge and should be rephrased.

* Abstract: "co-variation" is mentioned, but it is not clear what data exactly this refers to from the abstract only.

* Intro: "A more complete understanding of human cells requires a better understanding ..." - this could be rephrased.

* results: "Machine learning features from each experiment" - Machine learning uses features, it doesn't produce features (at least not in this context if I understood correctly). If features are actually produced and I misunderstood something, then more details are needed.

* results: the method "weighted matrix model approach (WMM)" is mentioned without explanation - A one-sentence description of the idea behind this approach could be given.

* P.13 "Since protein co-expression networks are compiled from separate datasets than datasets used in our construction of a human protein complex map," this sentence is not clear.

* Fig 7B: there is a typo in "not mutually exclusive"

We would like to first thank the reviewers for their constructive and supportive comments. We strongly believe their comments have helped us improve our manuscript.

Reviewer #1:

The paper "hu.MAP3.0: Atlas of human protein complexes by integration of >25,000 proteomic experiments" presents a comprehensive map of human protein complexes derived from the integration of over 25,000 mass spectrometry experiments using advanced machine learning workflows. This resource identifies more than 15,000 protein complexes, encompassing approximately 75% of the human proteome. The study highlights the accuracy and scope of hu.MAP3.0 compared to prior maps and provides novel insights, including the identification of co-varying complexes, mutually exclusive subunits, and functional predictions for understudied proteins. The dataset is enriched with structural insights from AlphaFold modeling and is made publicly accessible via EMBL-EBI's Complex Portal and a dedicated web interface. This work offers a valuable tool for understanding protein interactions and their implications in biology and disease.

One of the major achievements is the use of AF2-multimer to model mutually exclusive interaction partners - this is really novel.

Another major achievement is the use of co-variation both within and between complexes.

In summary this paper presents the state of art data for protein complexes. However, it is still not perfect as for instance JUNO-IZUMO1 is missing.

We greatly appreciate the reviewer's acknowledgement of the novelty and achievement of the work. The reviewer brings up a good point that the resource is still not complete. Within this work we show that hu.MAP3.0 is able to place ~70% of proteins within the human proteome in the context of protein complexes. Currently, as the reviewer points out, our complex map does not cover all interactions. Here we are limited by the data that has been collected in high throughput experiments which many, but not all, are from cell lines. Many PPIs are exclusive to specific tissues or cellular contexts that have not been experimentally characterized. For example, the JUNO-IZUMO1 interaction is specific to the egg and sperm interface, which the field has yet to analyze in high throughput. We now address the topic of coverage in the 2nd paragraph of the discussion. We look forward to greater coverage in our resource including context and cell type specific interactions as more experimental datasets become available.

I could reach humap3.proteomcomplexes.org the other day but today it is unreachable. Perhaps it is a good idea to check the stability of the website. This also meant that I could not check how easy it was to download or use the data (also humap2.proteincomplexes.org was only reachable via HTTP not HTTPS the other day - this should be fixed)

We recently migrated our hu.MAP webserver to Amazon AWS which we hope will provide better stability than our previous platform. Our migration is still ongoing for legacy web servers such as

hu.MAP2.0 but we expect that transition to be complete shortly. This will update all web servers to https. Note, hu.MAP3.0 is currently an https url. The url for the hu.MAP3.0 website is <https://humap3.proteincomplexes.org/>. We certainly would like to know if there are any connectivity issues in the future so that we may fix them promptly. Additionally, all of our identified complexes are mirrored on EBI's ComplexPortal web resource providing an additional point of access.

I would urge the authors to make all data FAIR compliant - not clear if it is - in particular ProteomeHD.2 scores are only available upon request - this is not OK.

We certainly understand the importance of accessible published data as hu.MAP is built on such data. We have uploaded the ProteomeHD.2 scores onto Mendeley Data, where they are now permanently and publicly accessible (<https://data.mendeley.com/datasets/76hjnczz7/2>; doi number: 10.17632/76hjnczz7.2). The Mendeley repository also contains the relevant documentation to facilitate the re-use of the covariation data while we finalise the ProteomeHD.2 manuscript, which will refer to the same Mendeley repository. Additionally we have updated the data availability section of the manuscript to include it.

In addition, we have created a HuggingFace dataset repository for the hu.MAP3.0 feature matrices and test and training sets (https://huggingface.co/datasets/DrewLab/hu.MAP_3.0). This is in addition to our downloads page on the hu.MAP3.0 website (<https://humap3.proteincomplexes.org/download>). This ensures all of our data is findable, accessible, interoperable, and reusable.

PS. The submission of manuscripts with all the figures at the end really makes it much harder for this reviewer to read it. This is something the journal should not allow. For every time a figure is mentioned I need to scroll down 20 pages to find the legend and then another 20 pages to find the figure, and finally try to find my way back. In the future I will not review papers formatted in this way.

We understand and provide a separate document that is reformatted to have figures inline.

Reviewer #2:

In this manuscript Fischer and colleagues describe the integration of a very large compendium of proteomics datasets, to generate a high confidence human protein-protein interaction network and list of protein complexes. This is an expansion of the previous studies done in 2017 (Drew et al. MSB 2017) and 2021 (Drew et al. 2021 MSB). The previous versions of this work have been very useful to the scientific community as it can be seen by the number of citations these papers have accumulated. The increase in data incorporated is primarily driven by the addition of the BioPlex 3 project dataset. Despite it being the addition of a single study it corresponds to a very large number of new experiments on its own. Therefore, this corresponds to a very useful integration effort that produces what is likely to be the current largest high confidence integrated dataset of human protein interactions and complexes.

The authors study this resource to provide, for example, functional annotation to understudied proteins as well for comparing it with a very large protein covariation analysis (ProteomeHD.2). This dataset is not yet published but is briefly described in the methods as consisting of co-abundance based estimates of protein associations derived from the reanalysis of nearly 2500 perturbation experiments done in human cells. Given the importance of this covariation dataset for the study that is here under review, I think this data would need to be made available in some form by the time of publication. The co-abundance analysis was used to find cases where proteins complex subunits do not covary, which includes examples of complexes that are mutually exclusive. The authors study mutual exclusivity in more detail by incorporating predicted structural models and in addition study the patterns of co-abundance of proteins in cancer cell proteomics datasets.

In summary, I think the work is a very useful data integration effort that creates an useful resource of a large community of scientists interested in human protein interactions. The methods are clear and data and predictions are readily available for re-use. The combined analysis of protein complexes and co-variation is very interesting as well as the study of mutual exclusive protein complexes. I have only a series of minor concerns that I hope the authors could consider:

1 - The ProteomeHD.2 dataset is critical for the work described here and anyone trying to reproduce the results would need to have access to it. I understand that there is a considerable amount of work behind ProteomeHD.2 but the authors would need to find a way to make that dataset available before publication.

We agree that this is important for transparency and accessibility and have made the ProteomeHD.2 scores available on Mendeley Data, where they are now permanently and publicly accessible (<https://data.mendeley.com/datasets/76hjnczz7/2>; doi number: 10.17632/76hjnczz7.2). The Mendeley repository also contains the relevant documentation to facilitate the re-use of the covariation data while we finalise the ProteomeHD.2 manuscript, which will refer to the same Mendeley repository. Additionally we have updated the data availability section of the manuscript to include it.

2 - The procedure for the machine-learning model for data-integration is well described but it is not clear in the end if there is any regularization involved or if the final model ends up using all features. The authors could provide as well a list of features with some model weight estimates so that it is clear which features end up being most useful in end result.

During the construction of our hu.MAP3.0 model we did not perform any explicit regularization or feature engineering. Autogluon, however, does filter features deemed to have no predictive value. In our application, there were five features that were determined by AutoGluon to have no predictive value during training: 'AvgSpec', 'AvgP', 'MaxP', 'Fold_Change', 'BFDR'. These features were ultimately not used for training and we have updated the manuscript text to reflect this. These features are derived from proximity labeling experiments and although they were

determined to have no predictive value, other proximity labeling features and weighted matrix model features of proximity labeling experiments are still found to be valuable at discriminating true co-complex interactions from false ones. Notably, some of these features are in the top 20 of feature importance (see response below on feature importance). All features are still provided within the feature matrix that is available for download but they are removed during automated model selection and therefore not used by the final model to make predictions.

Furthermore, as the reviewer suggests, understanding which features provide the greatest performance is beneficial for determining which datasets are most useful for PPI prediction. We now include a value analysis of our features, datasets, and methods in section "Feature Importance Analysis", Fig EV1D - EV1E, and Table EV8. While most features add value to the model, we see substantial importance for the Bioplex dataset as well as the Weighted Matrix Model method in predicting protein interactions.

3 - A very large number of complexes ends up not being annotated with any gene set. Is this mostly to do with the fact that most complexes are very small ? If the authors exclude the complexes that are <3 or <4 subunits, what is the fraction of non-annotated complexes ? Are there large >10 subunit complexes that do not have any predicted functional annotation ? Related to this, it would be useful to mention briefly the size distribution of complexes in the main text (shown in Figure S2).

There are a number of factors why we believe complexes may not be enriched for specific annotations. To determine if the size of the complex is a factor, we evaluated the distribution of complex size of all complexes vs enriched complexes (Fig EV2C) and do not see a substantial difference in the distributions. Alternative reasons why complexes are not enriched may be that annotation efforts are incomplete, annotation enrichment scores for complexes were not considered significant by our strict threshold, or we are identifying novel complexes with uncharacterized functions. Overall we observe a ~15 fold increase in enrichment compared to randomly generated gene sets of the same size. This is consistent with our previous observed enrichment of complexes in hu.MAP2.0. We have updated the main text of the manuscript to briefly mention the size distribution of complexes and illustrate the point with an example (i.e. MCM complex).

4 - I was surprised that the protein complex confidence didn't correlate much with the co-abundance data. It would be useful to also compare directly the protein-protein interaction scores. Is there a general correlation between protein-protein interaction confidence and protein covariation scores ? At the complex level, instead of confidence bin as in Fig S4, what is the correlation between average interaction confidence (for a given complex) and the corresponding average covariation score ?

We also initially expected higher correlation between co-abundance data and protein complex confidence but low correlation has been observed previously. There are well reasoned arguments for why this may be the case (Matalon et al., 2014). In particular, subunits of a complex may have additional functions outside of the complex driving its expression patterns.

Alternatively, a subunit may have weak affinity interaction in the complex and to compensate the subunit requires a higher expression level. We have updated the manuscript to point to these possibilities. We'd also like to thank reviewer #3 for pointing us to this work.

In regards to the correlation of hu.MAP3.0 and ProHD.2 scores, the distributions of both values are centered around 0 due to the vast number of protein pairs that do not physically interact or co-express. To correct for this and quantify the degree of correlation between the two scores we therefore set a threshold of 0.2 for both scores which resulted in a Pearson correlation coefficient of 0.29. The Pearson correlation coefficient between the average hu.MAP3.0 score and average ProHD.2 score on the complex level was low as well, 0.1. Both of these results agree with the previous observation of low correlation between complex subunit stoichiometry and expression levels (Matalon et al., 2014).

5 - Regarding the structural analysis and mutually exclusivity, which I really enjoyed, there is a much larger dataset of protein structure predictions in one of the cited studies (Jänes et al. *bioRxiv* 2024). Why not use those models as well to greatly expand the coverage ?

Thank you for this suggestion. We now analyze the much larger dataset from Jänes et al. We identify 5,991 mutually exclusive pairs and 3,095 structurally consistent pairs in total. This was a considerable increase over our original analysis. We also see similar trends in terms of co-expression being lower in mutually exclusive pairs than structurally consistent ones. Although expression is still the dominant way cells relieve the conflict of mutual exclusion, we do now see a slight statistically significant increase in the use of subcellular localization for relieving the conflict as well (Fig EV5C). We have updated the manuscript to reflect this updated analysis.

6 - Looking through the scored protein pairs in Table S7, I find around 160 thousand pairs, some of which have very low probabilities (0.0001). The manuscript mentions that the data covers 13,769 human proteins so the space of possible pairs among the proteins with data is many orders of magnitude larger than the 160k values. Figure 1C shows the number of interactions in humap3 as those 160k pairs but so I assume the Table S7 to be the correct number of defined interactions. Are the probability values in Table S7 correct ? What is the relationship between the probability value and model accuracy etc ?

Thank you for giving us the opportunity to clarify this point. The ~160k pairs are those that were predicted by our model and processed by our clustering procedure. Proteins within low scoring pairs have high scores with other proteins within the same complex. In other words, some pairs within the complex may not have confident support (i.e., low pairwise score), however, each subunit will have confident support (i.e. high pairwise score) with other proteins within the complex. As an example, PARD3-PARD6A (confidence score=0.003) has little supported data for their interaction in our model but is known to be co-complex in the PAR polarity complex. hu.MAP3.0 captures the PARD3-PARD6A interaction through their shared high confidence interactions with PRKCI. Based on our test set, we calculate 10% of these low confidence pairs that cluster together are positive pairs but just lacked evidence in our model. Further we observe the confidence of post clustering interactions increases where a protein pair with a

confidence score of 0.1 has a test set precision of >0.6 (Figure EV2D). We therefore made the choice to include these clustered pairs in Table EV7 (was Table S7) while reporting the confidence score of the interaction. We have updated the discussion to reflect this.

7 - Although this is a subjective opinion, I felt that there were parts of the manuscript that could be shortened without impacting on the messages. This includes: some of initial ML model building and clustering; the examples in "Identification of mutually exclusive subunits in hu.MAP3.0".

We have streamlined both sections that describe ML model building and the examples of the mutually exclusive subunits.

8 - Some the methods are described as running some scripts (Reduce redundancy of benchmark complexes and Generating test and train sets). This should be instead the description of the algorithm/steps that are performed by those scripts.

We kindly thank the reviewer for their comment on clarity regarding our methods as we ultimately want our tools to be accessible and used by other researchers. In addition to the examples of command-line implementation, we now provide more complete descriptions of the methodology behind the scripts used to generate the test and training benchmarks.

Reviewer #3:

Fischer et al. characterize the complexome of the human proteome. The work is an extension of their previous work Hu.Map2. Hu.Map3 increases both the number of complexes characterized (15,000 against 7,000) and the coverage of the proteome (13,800 vs 10,000 proteins). As in their previous work, a machine learning framework is used to score interactions from thousands of MS-based pull-down and co-fractionation experiments as well as from other resources such as proximity information. The integration of the BioPlex3 resource also serves as an important source of information.

The work builds on a solid resource and extends it further. It will be of immense value to the community. It also adds a novel perspective with a comparison between physical interactions and co-expression information, as inferred from ProteomeHD.2. The original analyses of mutually exclusive subunits and cancer-associated subunits also illustrate ways in which this resource can be used.

Several points could nevertheless be improved, in particular with respect to the presentation of the results and the mutually exclusive subunits, as described in the comments below.

Comments:

1. P.2: This final step results in 15,326 complexes covering 13,769 human proteins and consisting of 159,451 total scored protein interactions. These numbers indicate redundancy among complexes, and it would be valuable to provide some ideas/statistics regarding this

redundancy, e.g., the distribution of the number of complexes a protein appears in, and what these complexes correspond to biologically.

As the reviewer points out, there is a degree of redundancy within hu.MAP3.0 complexes. We now provide a plot showing the distribution of complexes per protein and show most proteins participate in multiple complexes (Fig EV2C). We note a partial reason for redundancy is protein complexes have a hierarchical nature where subcomplexes combine to form larger assemblies. We added text to the manuscript to describe the example of the MCM complex in which we identify a dimer subcomplex, the full MCM complex, as well as a supercomplex of MCM, GINS complex, and other replication factors.

2. P.4 and Fig S2: According to the plot, the larger the complexes are, the less reliable they become. The authors state "This is likely due to core members of complexes being confidently identified in high confidence levels and additional auxiliary subunits added to complexes in lower confidence clusters." This could be simply tested/analyzed.

We considered testing this in several ways but unfortunately we are not aware of a well curated database of core vs peripheral complex members or subcomplex vs full assemblies outside of a handful of examples. We therefore have rephrased this statement to reflect our clustering methods attempt to capture this behavior. We additionally provide the example of the MCM complex (described in the response to point above) having different levels of hierarchy.

3. P.22 "Additionally, we ensured that AlphaFold2 models had <5 overlapping residues between protein chains." I assume that the authors are referring to residue clashes. As the wording "overlapping residues" is also used by the authors to refer to "shared interfaces residues", this paragraph is not clear and should be re-written/better explained.

Thank you for the suggestion we have now clarified the description of our analysis of AlphaFold2 models for mutually exclusive interactions. We now consistently use the phrase "clashing residues" to describe residues that occupy the same 3D space, with the specific context clarified in each instance. We currently use 'overlap' only to describe the situation where shared interface residues (i.e., overlapping interface residue coordinates) between the interface of one AlphaFold2 modeled dimer and another interface from an additional AlphaFold2 modeled dimer, where the two structures contain a shared protein subunit. Additionally, we have reorganized the structural modeling section into more distinct parts to group similar filtering steps and improve overall flow. All updates can be found in the 'Identification and characterization of mutually exclusive protein subunits' section of the methods.

4. P.22 "Interface residues between each protein were tested for overlap (residue atoms < 4.0 Å). Pairs with >10 overlapping interface residues were considered mutually exclusive. Pairs with 0 overlapping interface residues were considered structurally consistent. Due to expected error in AlphaFold2 models, we considered pairs with < 10 and > 0 overlapping interface residues inconclusive." Although this will probably not change the conclusions, it could make sense to

use a definition of overlap that is a function of interface size (e.g., >5 or 10%) rather than an absolute number (10 residues).

Our logic in choosing an absolute number was that, in theory, a single residue clash (or a small number) would likely disrupt a binding interface regardless of its size. Our choice of 10 was to allow some error in misaligned residues resulting from AF modeling. Using the percentage of interface size approach would require a considerable number of overlapping residues for large interfaces which we did not consider physically realistic. Future approaches that score the interface by including both absolute and percentage based calculations are worth investigating.

5. P.22 "We tracked residue overlap between unique protein chains, regardless of their presence at defined interfaces, to further filter the structurally consistent pairs. Overlapping could result from clashes in poorly modeled regions and does not necessarily indicate structural inconsistency. Therefore, we excluded structurally consistent pairs which had > 100 overlapping residues between their unique protein chains." - here as well, if the goal is to filter out clashing residues, it would be clearer to write it explicitly. It would also be more accurate to conduct the analyses on "high confidence regions", e.g., those with a pLDDT above a given cut-off for example. I do not see this as a very critical point as conclusions are unlikely to change, but it could improve the results.

We thank the reviewer for this comment as it became an important consideration as we integrated an additional dataset that included more than 136,000 AlphaFold2 structures. To filter the structurally consistent pairs, we now adopt an approach that accounts for the confidence of the regions of unique chains that clash. Our approach now takes into account the pLDDT of the clashing residues. We have updated the methods section ('Identification and characterization of mutually exclusive protein subunits') to reflect these changes.

6. Since this work builds on and improves on hu.MAP2, it would be valuable to add a section in the methods where the main differences in the pipeline are mentioned. This would summarize both methodological differences (cut-off values, data sources, number of experiments used as input, number of features used for classification/learning) as well as output differences (number of proteins, PPIs, complexes, etc).

We now provide a table in the discussion section to highlight the major differences and updates between hu.MAP2.0 and hu.MAP3.0. It also includes the output differences as requested.

7. Along the same lines, an important result of this work is the larger coverage with almost 4,000 additional proteins. Therefore, this specific set should be highlighted/described better in the figures, and it would give the readers a more precise idea of their characteristics. For instance, are we equally confident about those proteins' interactions? If not, where is the confidence range situated when compared to the rest of the dataset? (a distribution of confidence score for old vs newly added proteins could be shown), etc.

We appreciate the reviewer's comment and agree that the 4,000 proteins newly added to this analysis should be highlighted better. When examining the confidence scores of all protein pairs containing at least one of the newly added proteins we find that the distribution of those confidence scores closely matches the background distribution of confidence scores for all protein pairs (Fig EV1C). This suggests that we are equally confident of interactions containing the newly added proteins. Furthermore, we now address this analysis within the second paragraph of "Construction of machine learning model to identify protein interactions".

8. The limitations of the current approach are not explored or explained in sufficient detail. To help with this, it would be useful to add a panel characterizing (e.g., through GO term enrichment and specific examples) the FP (negatives with score >0.9) and FN (positives with score <0.1) that are seen in the barplot Figure S1D.

We now include a description of limitations of our approach in the discussion section. We analyze high confident false positives and observe many have similar GO terms (Appendix Figure S1). Specifically, we see several examples of complexes with multiple variants in this set including SWI/SNF complex, AP-1 complex, and CFIm complex. Substantial evidence supports some of these interactions while others are ruled out using our mutually exclusive analysis. Thus we propose high confidence negative pairs be prioritized for future annotation efforts. With respect to false negatives, we observe a lack of data in our model for several low confidence positive interactions but some are recovered by being present in the same complex through indirect interactions. This suggests our clustering method mitigates some of our model's deficiencies.

9. In relation to the statement "However, we find that the degree of complex covariation correlates only mildly with hu.MAP3.0 confidence levels (Fig S4)" - I would suggest citing the following review, which discusses this concept extensively:

<https://pubmed.ncbi.nlm.nih.gov/24997301/>

Yes - thank you. We now include a further discussion of the topic itself and include the reference. Also see response to Reviewer 2 point 4.

10. It appears that mutually exclusive pairs show weaker co-expression than non-mutually exclusive ones, as one could expect (Fig 6K,L). Later on, it is shown that homomers also bring information with respect to such mutually exclusive subunits (Fig. 7B), so it is not clear why homomer information is not included in the analysis presented in Fig. 6. Related to this point, I would suggest using a recent comprehensive resource on human homomers:

<https://pubmed.ncbi.nlm.nih.gov/38325366/> - It would also be appropriate to cite what is to my knowledge the first attempt at mapping PPIs onto similar vs distinct interfaces (<https://pubmed.ncbi.nlm.nih.gov/17185604/>)

We thank the reviewer for this suggestion. We did use the first resource (Schweke, H. *et al.*) and described its use in the methods. We now remove homomers in our mutually exclusive analysis and emphasize this as a filtering step in both the 'Workflow' (Figure 6) and describe

this in the methods section. We now reference Kim et al. in the discussion section of the manuscript.

11. A limit to the structural analysis should be noted: the pLDDT should ideally not be used to assess the quality of a complex because it reflects on subunit quality rather than interaction quality (the latter requires PAE information). I understand that the authors cannot use PAE-derived metrics to filter out models because the dataset they downloaded does not provide this info, but a sentence could be added to note and explain this discrepancy with other AlphaFold analyses in the paper where they used the ipTM score (readers may not understand why the ipTM is used in certain analyses and not others).

Thank you for this comment. While we do not use PAE to filter out models, we do use IF pLDDT and pDockQ to remove models that do not have confident residues at the interface. Like the reviewer states, PAE was not provided for these models. We also add a comment regarding the ipTM values in the methods section describing AlphaFold3 analysis and further specify the use of IF pLDDT and pDockQ in the discussion section when describing our use of sensible cutoffs.

Language / clarity related comments:

* Abstract. "Unfortunately, we lack the subunit composition for all human protein complexes" - this sentence can be interpreted as a COMPLETE lack of knowledge and should be rephrased.

Updated language from "all" to "many".

* Abstract: "co-variation" is mentioned, but it is not clear what data exactly this refers to from the abstract only.

Added "mass spectrometry based" to the description of protein co-variation.

* Intro: "A more complete understanding of human cells requires a better understanding ..." - this could be rephrased.

We have updated the sentence to: "To gain a more complete understanding of human cells, it is essential to identify and characterize these protein complexes."

* results: "Machine learning features from each experiment" - Machine learning uses features, it doesn't produce features (at least not in this context if I understood correctly). If features are actually produced and I misunderstood something, then more details are needed.

Updated language to "Features representing evidence from each experiment ..."

* results: the method "weighted matrix model approach (WMM)" is mentioned without explanation - A one-sentence description of the idea behind this approach could be given.

Updated description to "... using our weighted matrix model approach (WMM) which uses the hypergeometric test to identify pairs of proteins seen in these large datasets more often than random (see methods)."

* P.13 "Since protein co-expression networks are compiled from separate datasets than datasets used in our construction of a human protein complex map," this sentence is not clear.

We have rephrased the sentence: Moreover, the ProteomeHD.2 database is compiled from datasets independent from those used to construct hu.MAP3.0. Therefore, interactions that agree between the two networks are more confident as there are multiple independent lines of evidence for their existence.

* Fig 7B: there is a typo in "not mutually exclusive"

Fixed.

18th Apr 2025

Manuscript Number: MSB-2024-12769R

Title: hu.MAP3.0: Atlas of human protein complexes by integration of > 25,000 proteomic experiments

Author: Samantha Fischer

Erin Claussen

Savvas Kourtis

Sara Sdelci

Sandra Orchard

Henning Hermjakob

Georg Kustatscher

Kevin Drew

Dear Dr Drew,

Thank you for sending us your revised manuscript. We have now received feedback from the three reviewers who evaluated your study. As you will see below, the reviewers are overall satisfied with the performed revisions. Before we can formally accept the manuscript for publication, we would ask you to address some remaining issues listed below:

- the remaining minor concerns of Reviewer #3.

On a more editorial levels:

1. Please remove the Authors' contribution section from the manuscript file.

2. Please provide up to five key words.

3. Funding information: The grant number for National Science Foundation -NSF2314278 is missing from the manuscript file, which needs to be added.

4. Please add a "DISCLOSURE AND COMPETING INTERESTS STATEMENT" heading above the sentence "The authors declare that they have no conflict of interest."

5. EV datasets/tables: source file names, titles, legends and manuscript callouts all need to be updated to "Dataset EV1-EV10".

6. The references need to be formatted according to the Molecular Systems Biology style. Please list up to 10 co-authors of a paper before adding et al. in the reference list. Citations should be listed in alphabetical order.

7. Appendix:

- Appendix should contain a title page with "Appendix for + manuscript title" and a Table of Content with page numbers of listed items.

- nomenclature should be Appendix Figure Sx and Appendix Table Sx throughout manuscript and Appendix PDF

8. Please address the following issues related to figure legends:

- Please note that the exact p-values are not provided in the legends of figures 6K, L; 7B, EV5C

- Please note that the box plots need to be defined in terms of minima, maxima, centre, bounds of box and whiskers, and percentile in the legends of figures 7A, D

- Please note that the box plots need to be defined in terms of percentile, minima, maxima in the legends of figures 3C, 6K, L; 7B

- Please note that information related to n is missing in the legends of figures 3C, 6K, L; 7A, B, D; EV5 C

9. I have slightly modified synopsis text (see attached). Please let me know if it's fine as is, or if you'd like to suggest any further changes.

10. Section order should be corrected: Title page - Abstract & Keywords - Introduction - Results - Discussion - Methods - Data Availability - Acknowledgements - Disclosure and Competing Interests Statement - References - Figure Legends - Table(s) - Expanded View Figure Legends.

Please resubmit your revised manuscript online, with a covering letter listing amendments and responses to each point raised by the referees. Please resubmit the paper ****within one month**** and ideally as soon as possible.

When you resubmit your manuscript, please download our CHECKLIST (<https://bit.ly/EMBOPressAuthorChecklist>) and include

the completed form in your submission. *Please note* that the Author Checklist will be published alongside the paper as part of the transparent process (<https://www.embopress.org/page/journal/17444292/authorguide#transparentprocess>)

Click on the link below to submit your revised paper.

Kind regards,
Jingyi

Jingyi Hou, PhD
Senior Editor
Molecular Systems Biology

*** PLEASE NOTE *** As part of the EMBO Press transparent editorial process initiative (see our Editorial at <https://dx.doi.org/10.1038/msb.2010.72> , Molecular Systems Biology will publish online a Review Process File to accompany accepted manuscripts. When preparing your letter of response, please be aware that in the event of acceptance, your cover letter/point-by-point document will be included as part of this File, which will be available to the scientific community. More information about this initiative is available in our Instructions to Authors. If you have any questions about this initiative, please contact the editorial office (msb@embo.org).

Reviewer #1:

With the exception that the website is still not available for me, I have no further comments.

Reviewer #2:

The authors largely addressed my previous concerns. This will be a fantastic resource for cell biology and many other fields.

Reviewer #3:

The authors have well addressed the comments raised. There is only one comment left, and addressing it would enhance the clarity and ease-of-use of this resource.

Related to previous Q1: The new plots in Fig. EV2A and C show important and useful information. It becomes apparent that most proteins are part of 3 or more complexes in Humap3, with hundreds being part of even 10 or more complexes. Such redundancy can be useful for some analyses. However, most users would probably want access to a simple list where a default strategy to remove redundancy is adopted. This could be done at two different levels. For example:

A possibility to keep only the largest complex would be useful e.g., if the full list of complexes contains ABC, AB, and BC, then only ABC would be made available and AB+BC could be flagged as "subcomplexes".

A possibility to merge largely redundant complexes with some overlap cut-off. For example, ABCDEFG and BCDEFGH could be merged into one ABCDEFGH complex.

Such post-processing would facilitate using the data presented here.

Additionally, an "Appendix Figure S1" is referred to in the discussion but it's not clear what figure it is.

We first want to thank the reviewers for their helpful and constructive comments. Our responses are inline below.

Reviewer #1:

With the exception that the website is still not available for me, I have no further comments.

We have checked the availability of the website and have not witnessed any problems. We will continue to monitor the uptime to insure access to the website (<https://humap3.proteincomplexes.org/>).

Reviewer #2:

The authors largely addressed my previous concerns. This will be a fantastic resource for cell biology and many other fields.

We thank the reviewer for their encouraging comments.

Reviewer #3:

The authors have well addressed the comments raised. There is only one comment left, and addressing it would enhance the clarity and ease-of-use of this resource.

Related to previous Q1: The new plots in Fig. EV2A and C show important and useful information. It becomes apparent that most proteins are part of 3 or more complexes in Humap3, with hundreds being part of even 10 or more complexes. Such redundancy can be useful for some analyses. However, most users would probably want access to a simple list where a default strategy to remove redundancy is adopted. This could be done at two different levels. For example:

A possibility to keep only the largest complex would be useful e.g., if the full list of complexes contains ABC, AB, and BC, then only ABC would be made available and AB+BC could be flagged as "subcomplexes".

A possibility to merge largely redundant complexes with some overlap cut-off. For example, ABCDEFG and BCDEFGH could be merged into one ABCDEFGH complex.

Such post-processing would facilitate using the data presented here.

We thank the reviewer for this comment to make our complexes more accessible to other researchers within the community. To generate a reduced set of complexes, we evaluated redundancy by calculating pairwise Jaccard similarity coefficients for all complexes, which we now describe at the end of "clustering and parameter selection" section of the methods and mention in the main text at the end of "Identification of protein complexes within protein interaction network". We chose a Jaccard coefficient that maintains similar k -cliques weighted precision and recall of the entire union set. Since k -cliques evaluation examines the underlying substructures of complexes it is not sensitive to redundancy, and similar performance between the full union set and the reduced set suggests redundancy has been removed without

substantial loss of performance for this set. We provide this reduced set for the convenience of the users in EV Dataset 4. This set contains 11,897 complexes, compared to the entire set of 15,326 complexes.

Additionally, an "Appendix Figure S1" is referred to in the discussion but it's not clear what figure it is.

We have now updated the appendix to include a cover page and table of contents. Appendix Figure S1 is located on page 2 and describes that covariation of hu.MAP3.0 complexes does not correlate with the determined confidence level of the complex. This is shown by comparison of median complex covariation of all complexes within each hu.MAP3.0 confidence level.

Editorial levels:

1. Please remove the Authors' contribution section from the manuscript file.

This section has been removed.

2. Please provide up to five key words.

Protein complex, protein interaction, mutually exclusive, disease candidates, machine learning

3. Funding information: The grant number for National Science Foundation -NSF2314278 is missing from the manuscript file, which needs to be added.

NSF231427 has been added to the Acknowledgements section.

4. Please add a "DISCLOSURE AND COMPETING INTERESTS STATEMENT" heading above the sentence "The authors declare that they have no conflict of interest."

This statement has been added.

5. EV datasets/tables: source file names, titles, legends and manuscript callouts all need to be updated to "Dataset EV1-EV10" .

We have renamed all supplementary tables as EV Datasets and updated their files names and references within the text, legends, and callouts.

6. The references need to be formatted according to the Molecular Systems Biology style. Please list up to 10 co-authors of a paper before adding et al. in the reference list. Citations should be listed in alphabetical order.

We have formatted the references in Molecular Systems Biology style.

7. Appendix:

- Appendix should contain a title page with "Appendix for + manuscript title" and a Table of Content with page numbers of listed items.

- nomenclature should be Appendix Figure Sx and Appendix Table Sx throughout manuscript and Appendix PDF

We have added a cover page and table of contents for the Appendix. We have also updated the references to the appendix figures with the proper nomenclature.

8. Please address the following issues related to figure legends:

- Please note that the exact p-values are not provided in the legends of figures 6K, L; 7B, EV5C
- Please note that the box plots need to be defined in terms of minima, maxima, centre, bounds of box and whiskers, and percentile in the legends of figures 7A, D
- Please note that the box plots need to be defined in terms of percentile, minima, maxima in the legends of figures 3C, 6K, L; 7B
- Please note that information related to n is missing in the legends of figures 3C, 6K, L; 7A, B, D; EV5 C

We have made all requested updates to the legends, including: adding N values to figures 3C; 6K, L; 7A, B, D; and EV5C, updating the boxplot descriptions for figures 3C; 6K, L; and 7A, B, D, and adding exact p-values to figures 6K, L; 7B, and EV5C.

9. I have slightly modified synopsis text (see attached). Please let me know if it's fine as is, or if you'd like to suggest any further changes.

These changes are fine.

10. Section order should be corrected: Title page - Abstract & Keywords - Introduction - Results - Discussion - Methods - Data Availability - Acknowledgements - Disclosure and Competing Interests Statement - References - Figure Legends - Table(s) - Expanded View Figure Legends.

We have updated the order of the manuscript sections.

9th May 2025

Manuscript number: MSB-2024-12769RR

Title: hu.MAP3.0: Atlas of human protein complexes by integration of > 25,000 proteomic experiments

Dear Dr Drew,

Thank you again for sending us your revised manuscript. We are now satisfied with the modifications made and I am pleased to inform you that your paper has been accepted for publication.

Sincerely,
Jingyi

Jingyi Hou, PhD
Senior Editor
Molecular Systems Biology
